



# New insights into the atmospheric mercury cycling in Central Antarctica and implications at a continental scale

**Hélène Angot[1], Olivier Magand[2, 1], Detlev Helmig[3], Philippe Ricaud[4], Boris Quennehen[2, 1], Hubert Gallée[2, 1], Massimo Del Guasta[5], Francesca Sprovieri[6], Nicola Pirrone[7], Joël Savarino[2, 1], Aurélien Dommergue[1, 2]**

[1]Univ. Grenoble Alpes, Laboratoire de Glaciologie et Géophysique de l'Environnement (LGGE), 38041 Grenoble, France.

[2]CNRS, Laboratoire de Glaciologie et Géophysique de l'Environnement (LGGE), 38041 Grenoble, France.

[3]Institute of Arctic and Alpine Research (INSTAAR), University of Colorado, Boulder, CO 80309-0450, USA.

[4]CNRM/GAME, Météo-France/CNRS, 42 avenue de Coriolis, 31057 Toulouse, France.

[5]CNR-Istituto Nazionale di Ottica, Largo E. Fermi 6, Firenze, 50125, Italy.

[6]CNR-Institute of Atmospheric Pollution Research, Division of Rende, Italy.

[7]CNR-Institute of Atmospheric Pollution Research, Montelibretti, Rome, Italy.

Correspondence to: A. Dommergue (aurelien.dommergue@ujf-grenoble.fr)

**Abstract**

Under the framework of the GMOS project (Global Mercury Observation System) atmospheric mercury monitoring has been implemented at Concordia Station on the high-altitude Antarctic plateau (75°06'S, 123°20'E, 3220 m above sea level). We report here the first year-round measurements of gaseous elemental mercury (Hg(0)) in the atmosphere and in snowpack interstitial air on the East Antarctic ice sheet. This unique dataset shows evidence of a continuous oxidation of atmospheric Hg(0) in summer (24-hour daylight) due to the high oxidative capacity of the Antarctic plateau atmosphere at this period of the year. Summertime Hg(0) concentrations exhibited a pronounced daily cycle in ambient air with maximal concentrations around midday. Photochemical reactions and chemical exchange at the air/snow interface were prominent, highlighting the role of the snowpack on the



atmospheric mercury cycle. Our observations reveal a 20 to 30% decrease of atmospheric
Hg(0) concentrations from May to mid-August (winter, 24-h darkness). This phenomenon has
never been observed elsewhere and likely results from a gas-phase oxidation and/or
heterogeneous reactions. We also reveal the occurrence of multi-day to weeklong atmospheric
Hg(0) depletion events in summer, not associated with depletions of ozone, and likely due to a
stagnation of air masses above the plateau triggering an accumulation of oxidants within the
shallow boundary layer. Our observations suggest that the inland atmospheric reservoir is
depleted in Hg(0) in summer. Due to katabatic winds flowing out from the Antarctic plateau
down the steep vertical drops along the coast and according to observations at coastal
Antarctic stations, the striking reactivity observed on the plateau most likely influences the
cycle of atmospheric mercury at a continental scale.
**1   Introduction**
Mercury biomagnifies in its methylated form in aquatic food webs to elevated levels in
freshwater and marine fish, causing adverse health effects to wildlife and humans (Mason et
al., 2012). In 2013, the Minamata Convention (UNEP, 2013) was adopted and opened for
signature to reduce the exposure of populations to this worldwide contamination. Gaseous
elemental mercury (Hg(0)), the most abundant form of mercury in the atmosphere, is
efficiently transported around the globe, and even remote areas receive significant inputs of
anthropogenic mercury by long-range atmospheric transport, as recently reported in modeling
and observational studies (Weiss-Penzias et al., 2007;  Lin et al., 2010).
Hg(0) can be oxidized into highly reactive and water-soluble gaseous and/or particulate
divalent species (Hg(II) and Hg(p), respectively) (Lin and Pehkonen, 1999) leading to the
formation and subsequent rapid deposition of reactive mercury onto environmental surfaces
(Hedgecock and Pirrone, 2004). Upon deposition mercury can be reemitted back to the
atmosphere or may enter the food chain through the conversion of Hg(II) to its methylated
form (Driscoll et al., 2013). Effects and toxicity of mercury depends on this complex cycle,
which is still not fully understood, and are only indirectly related to regional and global
emissions (Driscoll et al., 2013). A better understanding of atmospheric mercury chemistry
will lead to improved global transport and deposition models and could help refine pollution-
control strategies around the world.





New oxidation pathways, discovered in 1995 in the Arctic (Schroeder et al., 1998) and
highlighting the influence of halogen radicals on Hg(0) oxidation in spring, changed our
understanding of the mercury cycle. While the Arctic has been extensively monitored, there is
still much to be learned from the Antarctic continent where studies are scarce (Dommergue et
al., 2010), especially on the high altitude plateau (see Fig. 1). The Antarctic plateau – ice-
covered area of ~ 7 million km$^2$ – was first considered to be chemically-inactive and a giant
cold trap (Eisele et al., 2008) for atmospheric species, including mercury. It turned out to be
highly photochemically active (Davis et al., 2001; Grannas et al., 2007) during the sunlit
period with oxidant concentrations approaching those of tropical or urban mid-latitude
environments (Eisele et al., 2008; Kukui et al., 2014). Earlier studies (Brooks et al., 2008;
Dommergue et al., 2012) – the only two mercury studies ever carried out on the high-altitude
Antarctic plateau with modern instruments – also suggested, based on short-term observations
(a few weeks) in summer, an intense reactivity of mercury on the plateau at the air/snow
interface. In this context, and under the framework of the GMOS project (Global Mercury
Observation System, www.gmos.eu), atmospheric mercury was continuously monitored at
Concordia Station (see Fig. 1) since 2012 and, for the first time, Hg(0) has been monitored
year-round in both the snow interstitial air and the overlying atmosphere in 2013. Given harsh
weather conditions (see section 2.1), and technical and logistical limitations, presenting such a
record is in itself an important achievement. The main objective of this study is to provide
new insights into the year-round cycling of gaseous mercury on the Antarctic plateau.

**2 Experimental Section**
**2.1 Sampling site**
Year-round measurements of gaseous mercury were conducted in 2012 and 2013 at the
French/Italian Concordia Station (75°06'S, 123°20'E, 3220 m above sea level), located on the
Antarctic plateau, 1100 km away from the nearest coast of East Antarctica (see Fig. 1).
Concordia Station is a regional topographic maximum on the plateau; the surface terrain
slopes do not exceed 1% (Genthon et al., 2010). The air temperature ranges between -20 °C in
summer and -70 °C in winter, with an annual mean value of -45 °C (Pietroni et al., 2012).
There is permanent daylight in summer and permanent darkness in winter. Snow
accumulation is ~ 10 cm/year (Genthon et al., 2013).





## 2.2 Sampling instrumentation

Instrumentation was located in a below-surface shelter at the edge of the "clean area", 800 m away from the main camp and upwind with respect to the dominant wind direction (south west). In 2012, year-round measurements were performed in the atmospheric boundary layer at about 500 cm above the snow surface. In 2013, measurements were performed in both the atmosphere and in snowpack interstitial air for several trace gases including gaseous mercury and ozone ($O_3$). Sampling instrumentation included one 10 m meteorological tower for above-surface gradient sampling and two multi-inlet snow sampling manifolds ("snow towers") for measuring trace gases at various depths in interstitial air (Bocquet et al., 2007; Seok et al., 2009). The 10 m meteorological tower was installed ~ 15 m upwind of the underground instrument shelter. It accommodated three gas inlets at 1070 cm, 210 cm, and 25 cm above the snow surface (see Fig. 2a). Trace gas measurements were acquired on each snow tower at six height levels: 50 and 10 cm above the snow surface, and 10, 30, 50, and 70 cm below the snow surface (see Fig. 2b). Sampling lines were purged continuously at 5 L/min on the meteorological tower and intermittently at ~ 2-3 L/min on the snow towers. On each snow tower, inlets were fitted with a small glass fibre filter in PTFE housing (25 mm Acrodisc syringe filters, Pall Life Sciences, Ann Arbor, Michigan, USA) to prevent snow crystals from entering the PFA sampling lines. Sampling lines were inside insulation tubing and the temperature of the sampling lines was maintained at a level 5-10 °C warmer than the snowpack temperature with a heat trace to prevent water vapor from freezing and clogging the lines. An automatic sampling pattern was implemented: trace gases were collected sequentially from the uppermost inlets on the meteorological tower to deepest levels of the snow towers. Measurements were taken for 10 min from each inlet.

## 2.3 Gaseous mercury measurements

Measurements were performed using a Tekran 2537A analyzer (Tekran Inc., Toronto, Canada) based on the amalgamation of mercury onto a gold cartridge followed by thermal desorption and detection by an integrated cold vapor atomic fluorescence spectrometer (CVAFS) at 253.7 nm (Fitzgerald and Gill, 1979; Bloom and Fitzgerald, 1988). The presence of two gold cartridges allowed alternating sampling and desorption modes and thus a continuous analysis in the pre-filtered (0.45 µm PTFE filter) and unheated sample air stream. The sampling resolution was 5 min with a sampling flow rate of 0.8 L/min. Concentrations are expressed in nanograms per cubic meter at standard temperature and pressure (273.15 K,



1013.25 hPa). Using both a 0.45 µm PTFE filter at the entrance inlet of the sample line, and
an unheated ¼"PTFE sample line, we assume that only Hg(0) (vs. total gaseous mercury,
defined as the sum of gaseous mercury species) was efficiently collected and subsequently
analyzed in this study (Steffen et al., 2002; Temme et al., 2003; Steffen et al., 2008).
*Quality assurance and quality control procedures*
An automatic calibration step of the Tekran 2537A analyzer was carried out every 25 h with
an internal mercury permeation source. External calibrations were performed twice a year by
injecting manually saturated mercury vapor taken from a temperature-controlled vessel
(Tekran 2505 mercury vapor calibration unit, Hamilton digital syringe). As described by
Angot et al. (2014), fortnightly to monthly routine maintenance operations were performed. A
software programme was developed at the LGGE (Laboratoire de Glaciologie et Géophysique
de l'Environnement) in accordance with quality control practice in well-established North
American networks (Steffen et al., 2012). Based on various flagging criteria (Munthe et al.,
2011; D'Amore et al., 2015), it enabled rapid data processing in order to produce clean time
series of Hg(0). The detection limit is estimated at 0.10 ng/m$^3$ (Tekran, 2011).

### 2.4   Surface snow sampling and analysis

Surface snow samples (first cm) were collected weekly from February 2013 to January 2014
using acid cleaned PTFE bottles and clean sampling procedures. Upon collection, samples
were stored in the dark at -20 °C. Field blanks, carried out by opening and closing a bottle
containing mercury-free water, were regularly conducted. Surface snow samples and field
blanks were analyzed for total mercury using a Tekran Model 2600. Quality assurance and
quality control included the analysis of analytical blanks, replicates, internal standards, and
spiked materials. The limit of quantification – calculated as 10 times the standard deviation of
a set of 3 analytical blanks – amounted to 0.3 ng/L with a relative accuracy of ± 8%.

### 2.5   Ancillary parameters

*Ozone*
Measurements were performed using a UV absorption monitor (Thermo Electron
Corporation, Franklin, MA), model 49I in 2012 (Legrand et al., Submitted to ACP) and model
49C in 2013. The instruments were calibrated against the National Oceanic and Atmospheric
Administration Global Monitoring Division, Boulder, Colorado, standard.



*Air mass back trajectories*
Air mass back trajectories were computed using the Lagrangian model FLEXPART (Stohl et
al., 1998; Stohl and Thomson, 1999; Stohl et al., 2005) run in the backward mode and driven
by NCEP (National Center for Environmental Predictions) GFS (Global Forecast System)
final meteorological fields. Simulations were done every day at 1200 UTC in 2012 and 2013.
For each simulation, 20000 pseudo-particles were released by the model around the position
of Concordia Station and tracked for 5 days in three layers of altitude (0-0.1, 0.1-4 and 4-10
km above ground level). Simulations at an altitude of 4-10 km were computed in order to
investigate the potential occurrence of upper troposphere/lower stratosphere intrusions. For
each 1-h time step, the model produced a normalized particle residence time (in seconds)
within an output grid of 0.5x0.5°. The sum of the 5 days outputs provided potential emission
sensitivities (PES, in seconds) for the three layers of altitude. PES in a particular grid cell is
proportional to the particle residence time in that cell. It should be noted that, in Antarctica,
the meteorological data driving the FLEXPART transport model rely on sparse
measurements. Consequently, the trajectories calculated in this region are often associated
with relatively high uncertainties.
*Height of the boundary layer and shortwave radiation*
The height of the boundary layer and downwelling shortwave radiation were calculated by the
MAR regional atmospheric model (Modèle Atmosphérique Régional). MAR was developed
at the LGGE for Polar Regions and the simulations have been evaluated against
meteorological observations made at Concordia Station (Gallée and Gorodetskaya, 2010;
Gallée et al., 2015).
*Meteorological data*
Temperature, wind speed and direction were recorded at six height levels on a 45 m tower.
The general observation set up is described by Genthon et al. (2010).
*Ice precipitation*
A tropospheric depolarization LIDAR (Light Detection And Ranging) operating at 523 nm
provided tropospheric profiles of aerosol and clouds every 5 min allowing detection of
water/ice clouds, snow drift, diamond dust and pollution plumes.
*Tropospheric temperature and integrated water vapor*





A $H_2O$ Antarctica Microwave Stratospheric and Tropospheric Radiometers (HAMSTRAD)
instrument was used for the detection of the 60-GHz oxygen and the 183-GHz water vapor
lines allowing measurement of tropospheric temperature and water vapor profiles,
respectively, together with integrated water vapor (IWV) every 7 min. The instrument is fully
automated and a liquid nitrogen calibration is performed twice per year (Ricaud et al., 2015).
*Eddy diffusivity and friction velocity*
The Eddy diffusivity was calculated as follows (Xiao et al., 2014):
$K = k \, u_* \, z / \varphi_h$     (1)
where $k$ (set to 0.40) is the von Karman constant, $u_*$ the friction velocity (m/s), $z$ the
measurement height (m), and $\varphi_h$ the Obukhov stability function. According to Frey et al.
(2013), the stability function was $\varphi_h = 0.95 + 4.62 \frac{z}{L}$ for stable conditions above snow (King
and Anderson, 1994), and $\varphi_h = 0.95 \left(1 - 11.6 \frac{z}{L}\right)^{-0.5}$ for unstable conditions (Hoegstroem,
1988). $u_*$ and L (the Obukhov length (m)) were computed from the three-dimensional wind
components ($u$, $v$, $w$) and temperature measured by a Metek sonic anemometer mounted at 8
m above the snow surface.

**3 Results and Discussion**
**3.1 Seasonal variation of Hg(0) in ambient air**
The seasonal boundaries were defined according to the transitions in downwelling shortwave
radiation (see Fig. 3b) as follows: winter from May to mid-August, spring from mid-August
to October, summer from November to mid-February, and fall from mid-February to April.
The record of atmospheric Hg(0) over the entire 2012-2013 period is displayed in Fig. 3a.
Hg(0) concentrations ranged from below 0.10 to 2.30 ng/m³, with average values amounting
to 0.76 ± 0.24 ng/m³ in 2012, and to 0.81 ± 0.28 ng/m³, 0.84 ± 0.27 ng/m³, and 0.82 ± 0.26
ng/m³ in 2013 at 25, 210, and 1070 cm above the snow surface, respectively. No significant
difference was observed between annual averages of Hg(0) concentrations measured at the
three inlets of the meteorological tower in 2013 ($p$ value = $3.1.10^{-14}$, Mann-Whitney test).
These mean annual Hg(0) concentrations are lower than annual averages reported at coastal
Antarctic stations (i.e., 0.93 ± 0.19 ng/m³ for Hg(0) at Troll (Pfaffhuber et al., 2012) and 1.06
± 0.24 ng/m³ for total gaseous mercury at Neumayer (Ebinghaus et al., 2002)).





Unlike in winter, Hg(0) concentrations were highly variable during the sunlit period with
concentrations ranging from below 0.10 ng/m$^3$ to 1.50-2.00 ng/m$^3$, up to twice the average
background levels recorded in the Southern Hemisphere mid-latitudes (Slemr et al., 2015).
These seasonal features, in good agreement with observations at other Antarctic stations
(Ebinghaus et al., 2002; Pfaffhuber et al., 2012), suggest the existence of a photochemically-
induced reactivity of atmospheric mercury during the sunlit period.

## 3.2 Summertime continuous oxidation of Hg(0) in ambient air and Hg(II) deposition onto snowpack

In summer, the mean atmospheric Hg(0) concentration was 0.69 ± 0.35 ng/m$^3$. This means
that Hg(0) concentrations are ~ 25% lower than levels recorded at the same period of the year
at coastal Antarctic stations (Pfaffhuber et al., 2012; Ebinghaus et al., 2002; Sprovieri et al.,
2002). Total mercury concentrations in surface snow samples were highly variable (median
value: 4.8 ng/L, range: < detection limit – 73.8 ng/L, see Fig. 4) and were higher in summer
(median value: 10.4 ng/L, range: 1.3 – 73.8 ng/L), suggesting that divalent mercury species
were preferentially deposited onto the snowpack at this period of the year. The lower Hg(0)
concentrations in ambient air along with high total mercury concentrations in surface snow
samples suggest a continuous oxidation of Hg(0) in ambient air in summer, followed by the
deposition of oxidation products on surface snow. This hypothesis is further supported by
elevated oxidized mercury concentrations measured on the Antarctic plateau at South Pole in
summer (0.10 – 1.00 ng/m$^3$) by Brooks et al. (2008).
The oxidative capacity of the Antarctic plateau atmosphere is elevated in summer, as
evidenced by several studies (Davis et al., 2001; Grannas et al., 2007; Eisele et al., 2008;
Kukui et al., 2014), likely explaining this continuous oxidation of Hg(0) in ambient air.
Among these oxidants, NO$_2$, RO$_2$, and OH are particularly abundant at Concordia Station in
summer (Frey et al., 2013; Kukui et al., 2014) and a recent study provided as a first estimate
a BrO mixing ratio of 2-3 pptv near the ground during sunlight hours (Frey et al., 2015).
Given the current understanding of mercury oxidation and the lack of continuous halogens
measurements, we were not able to identify the exact mechanism for the reactivity observed at
Concordia Station. A two-step oxidation mechanism, favored at cold temperatures, is worth
being considered further. The initial recombination of Hg(0) and Br is followed by the
addition of a second radical (e.g., I, Cl, BrO, ClO, OH, NO$_2$, or HO$_2$) in competition with
thermal dissociation of the HgBr intermediate (Goodsite et al., 2004; Wang et al., 2014).



According to Dibble et al. (2012), $HO_2$, NO, $NO_2$, and $NO_3$ bound Hg(0) too weakly to
initiate its oxidation in the gas phase and reactions of the HgBr intermediate with $NO_2$, $HO_2$,
ClO, and BrO are more important than with Br and OH. Further modeling or laboratory
chamber studies investigating the fate of Hg(0) in the presence of various potential oxidants
are needed to improve our understanding of the mechanisms.

### 251   3.3  Hg(0)/Hg(II) redox conversions within the snowpack

Hg(0) moves easily between the atmosphere and the cryosphere (Durnford and Dastoor,
2011). A fraction of deposited mercury species can be reduced (the reducible pool) and
subsequently reemitted to the atmosphere as Hg(0). Reduced mercury can concurrently be
reoxidized within the snowpack. In Fig. 5, the redox processes occurring within the snowpack
are shown along with several other processes that govern mercury exchange at the air/snow
interface. These processes are discussed in details in the following sections.

### 258   3.3.1  Sunlit period

Fig. 6 depicts the mean Hg(0) concentration at various heights above and below the snow
surface (in the interstitial air of the snow) for all seasons. During the sunlit period (summer,
spring/fall), Hg(0) concentration peaked in the upper layers of the snowpack and then
decreased with depth, with levels in the snow interstitial air (SIA) dropping below
atmospheric values.
Hg(0) is generally produced in the upper layers of the snowpack – as the result of a
photolytically initiated reduction of Hg(II) (Lalonde et al., 2003) – and diffuses upward and
downward. According to our observations, Hg(0) concentration peaked at ~ 10 cm below the
snow surface (see Fig. 6). Similarly, Brooks et al. (2008) reported Hg(0) concentrations
peaking at a depth of 3 cm at South Pole. Below the top layer, the actinic flux decreases
exponentially with depth (King and Simpson, 2001; Domine et al., 2008). The light
penetration depth (*e*-folding depth) is the depth at which the actinic flux's magnitude is $1/e$ of
its incident value (Perovich, 2007). It is estimated that ~ 85% of the photoreduction occurs in
the top two *e*-folding depths (King and Simpson, 2001). At Concordia Station, the *e*-folding
depth is ~ 10 cm at 400 nm for the windpack layers (France et al., 2011), which supports our
observations. As previously mentioned, reduced mercury can concurrently be reoxidized
within the snowpack. Below the top layer, Hg(0) concentration in the SIA dropped with depth
(see Fig. 6) suggesting that oxidation dominated in the deepest layers – in good agreement



with observations within the snowpack at Kuujjuarapik/Whapmagoostui, Québec, Canada
(Dommergue et al., 2003) – leading to the formation of a Hg(II) reservoir.
The amount of Hg(0) emitted from the snowpack to the atmosphere depends on the balance of
reduction and oxidation processes within the upper layers of the snowpack (Durnford and
Dastoor, 2011). Fig. 7 depicts the hourly mean atmospheric and interstitial air Hg(0)
concentrations. Our observations indicate that summertime Hg(0) concentration in the upper
layers of the snowpack exhibited a diurnal cycle and peaked in the afternoon (see Fig. 7a,
lower panel). Conversely, in spring/fall, Hg(0) concentration reached a maximum at night and
a minimum near midday in the upper layers of the snowpack (see Fig. 7b). The balance of
reduction and oxidation processes within the upper layers of the snowpack suddenly shifted in
summer. Similarly, Faïn et al. (2008) found that reduction dominated during summer and
oxidation in spring in the upper layers of the snowpack at Summit, Greenland.
It is worth noting that Hg(0) concentration in the SIA was occasionally very high. For
instance, on 24 February 2013, Hg(0) concentration reached 3.00 ng/m$^3$ at a depth of 10 cm.
During this event, ice precipitation was observed at Concordia Station with depolarization
values greater than 30% (see Fig. 8). This suggests that the presence of ice crystals could
enhance the dry deposition of Hg(II) species onto the snow surface leading to increased Hg(0)
formation in the upper layers of the snowpack. Indeed, due to an elevated specific surface
area, mercury-capture efficiency of ice crystals is high (Douglas et al., 2008). Unfortunately,
due to a low sampling frequency of surface snow samples (weekly), total mercury
concentrations cannot be used to study further the relationship between the occurrence of ice
precipitation events and dry deposition of mercury species.

### 3.3.2  Winter

Contrarily to the sunlit period, Hg(0) concentration increased with depth in the SIA in winter
(see Fig. 6). The average (winter) Hg(0) concentration amounted to 3.60 ng/m$^3$ at a depth of
70 cm and was observed at a temperature of about -60 °C and not related to any change in
atmospheric composition. Our observations are in agreement with earlier studies indicating
that reduction of Hg(II) species is possible in the dark (Ferrari et al., 2004;  Faïn et al., 2007;
Ferrari et al., 2008). The production of Hg(0) might be due to the reduction of Hg(II) species
accumulated in the deepest layers of the snowpack during the sunlit period (see section 3.3.1).
This shift from oxidation to reduction in the deepest layers of the snowpack at the beginning
of winter remains unexplained.





### 3.4  Boundary layer dynamics and its influence on Hg(0) in ambient air


Several studies highlighted that the atmospheric turbulence at Concordia Station in summer
influences the vertical flux and concentration profiles of various atmospheric species
(Legrand et al., 2009; Dommergue et al., 2012; Kerbrat et al., 2012; Frey et al., 2013).


### 3.4.1  Summertime Hg(0) diurnal cycle


Based on a week of measurements made at Concordia Station in January 2009, Dommergue et
al. (2012) reported that atmospheric Hg(0) exhibited a significant and daily cycle with
maximal concentrations around noon. We show that this daily cycle occurred all along the
summer, with low atmospheric Hg(0) concentrations (~ 0.50 ng/m$^3$) when solar radiation was
minimum and a maximum (~ 0.80 ng/m$^3$) around noon (see Figs. 7a, upper panel and 9g).
Such a pronounced daily cycle has never been observed at other Antarctic stations
(Dommergue et al., 2010; Pfaffhuber et al., 2012). Several studies showed that Hg(0)
emission from the snowpack maximizes near midday (e.g., Steffen et al., 2002; Ferrari et al.,
2005; Brooks et al., 2006; Faïn et al., 2007; Ferrari et al., 2008; Johnson et al., 2008). As
suggested by Durnford and Dastoor (2011), the noon emission does not necessarily reflect
maximum concentrations of cryospheric Hg(0) around midday (Hg(0) concentration peaked
in the afternoon at 10 cm below the snow surface, see section 3.3.1) and could be driven by
ventilation generated by atmospheric thermal convection. Stable boundary layers are almost
ubiquitous in Polar Regions due to radiation cooling (Anderson and Neff, 2008). However,
convective boundary layers have been observed in summer at polar domes at Concordia
Station (King et al., 2006) and Summit in Greenland (Cohen et al., 2007). Fig. 9 displays the
hourly mean variation of several parameters. As illustrated by Figs. 9a and 9c, and in
agreement with earlier observations (e.g., Argentini et al., 2005; Pietroni et al., 2012;
Argentini et al., 2013), there was a strong diurnal cycle in near-surface temperature and wind
speed in summer at Concordia Station. These observations are typical for locations where a
convective boundary layer develops as a response to daytime heating (King et al., 2006), as
can be seen in Fig. 9d. In a convective boundary layer, vertical mixing is enhanced during
convective hours (Anderson and Neff, 2008), as shown in Figs. 9e and 9f by increasing values
for the Eddy diffusivity ($K$) and the friction velocity ($u_*$, indicative of the strength of the
mixing processes in the surface layer (Neff et al., 2008)).


In summary, the observed summertime Hg(0) diurnal cycle in ambient air might be due to a
combination of factors: i) a continuous oxidation of Hg(0) in ambient air due to the high






oxidative capacity on the plateau, ii) important Hg(II) deposition onto snowpack, and iii) important emission of Hg(0) from the snowpack during convective hours following photoreduction of Hg(II) in the upper layers of the snowpack.

### 3.4.2 Elevated Hg(0) concentrations in fall

In fall, Hg(0) concentrations in ambient air no longer peaked around midday (see Fig. 9g) and were in average 67% higher than during the summer, exceeding levels recorded at lower latitudes in the Southern Hemisphere (Slemr et al., 2015). At this period of the year, the boundary layer lowered to ~ 50 m in average and no longer exhibited a pronounced diurnal cycle (see Figs. 3c and 9d). We believe that the shallow boundary layer could cause Hg(0) concentrations in ambient air to build up to where they exceeded levels recorded at lower latitudes in the Southern Hemisphere because Hg(0) was dispersed into a reduced volume of air, limiting the dilution. Similarly, NOx mixing ratios are enhanced when the boundary layer is shallow (Neff et al., 2008; Frey et al., 2013). Elevated Hg(0) concentrations were also likely favored by the fact that oxidation in ambient air was weaker under lower solar radiation.

### 3.5 Multi-day depletion events of atmospheric Hg(0)

First discovered in the Arctic (Schroeder et al., 1998), atmospheric Hg(0) depletion events result from an oxidation by reactive bromine species released during springtime explosions in coastal regions (Durnford and Dastoor, 2011 and references therein) and are concurrent with tropospheric $O_3$ depletion events (Simpson et al., 2007). Despite the distance of Concordia Station from the coast (1100 km), a Hg(0) depletion event was observed on 11 September 2013 due to a maritime air transport event (see Fig. 10e). During this event, Hg(0) concentrations exhibited a strong positive correlation with $O_3$ mixing ratios (rho = 0.94, $p$ value = $5.10^{-7}$).

From 19 January to 8 February 2012, and from 5 to 20 February 2013, we, however, observed different depletion events. While atmospheric Hg(0) concentrations dropped and remained low ($0.39 \pm 0.19$ ng/m$^3$ from 19 January to 8 February 2012, $0.41 \pm 0.21$ ng/m$^3$ from 5 to 20 February 2013) for several weeks (see Figs. 3a, 11a, and 11e), $O_3$ showed no abnormal variability (see Figs. 11d and 11h). These depletion events occurred as air masses stagnated over the Antarctic plateau (see Figs. 10a and 10b) according to our FLEXPART simulations. This stagnation of air masses is confirmed in 2013 (see Figs. 11f and 11g) by a decrease of



temperature at 10 m a.g.l (from -29 ± 3 °C in January to -43 ± 4 °C during the Hg(0) depletion
event) and a low integrated water vapor (0.40 ± 0.13 kg/m$^2$ during the Hg(0) depletion event
vs. 0.77 ± 0.20 kg/m$^2$ in January). In both 2012 and 2013, depletions of Hg(0) ended when air
masses started moving out of the plateau (see Figs. 10c and 10d).
While previous studies attributed high Hg(II) concentrations in the Antarctic summer to
subsiding upper tropospheric air (Holmes et al., 2006; Brooks et al., 2008), potential emission
sensitivities suggest that these depletions of Hg(0) were unlikely concomitant with upper
troposphere/lower stratosphere intrusions (see Figs. 10a and 10b, PES at 4-10 km). This is
also confirmed by stable O$_3$ mixing ratios. High altitude vertical profiles of Hg(0) should be
carried out to rule out this hypothesis of subsiding upper tropospheric air. We suggest that
these Hg(0) depletion events observed at Concordia Station result from processes occurring
within the shallow boundary layer. Since O$_3$ was not depleted during these events, Hg(0)
depletion cannot be accounted for by bromine oxidation alone. FLEXPART simulations along
with integrated water vapor and temperature measurements indicate that these Hg(0) depletion
events occurred as air masses stagnated over the Antarctic plateau. As highlighted in section
3.2, the oxidative capacity is high in summer on the plateau (Davis et al., 2001; Grannas et
al., 2007; Eisele et al., 2008; Kukui et al., 2014). This air mass stagnation might favor an
accumulation of oxidants within the shallow boundary layer (< 300 m in average), leading to
an oxidation of Hg(0) stronger than usual.
### 3.6   Decreasing trend in winter
While stable concentrations were expected in winter given the absence of photochemistry, our
observations reveal a 20 to 30% decrease of atmospheric Hg(0) concentrations from May to
mid-August (see Fig. 3a). Conversely, Hg(0) concentrations remained stable at Troll and
Neumayer from late fall through winter (Ebinghaus et al., 2002; Pfaffhuber et al., 2012). This
decreasing trend observed in winter might be due to several mechanisms, including gas-phase
oxidation and heterogeneous reactions.
#### 3.6.1   Hypothesis on a gas-phase reaction
Several studies suggested the involvement of nitrate radicals in the night-time oxidation of
Hg(0) (Mao and Talbot, 2012; Peleg et al., 2015). However, as previously mentioned, Dibble
et al. (2012) indicated that NO$_3$ bound Hg(0) too weakly to initiate its oxidation in the gas
phase. Another potential oxidant is O$_3$, with this reactant reaching a maximum in the winter





(Legrand et al., 2009). However, according to some theoretical studies (e.g., Hynes et al.,
2009), reaction (R1) is unlikely to proceed as a homogeneous reaction. Several experimental
studies confirmed the major product of reaction (R1) to be solid mercuric oxide, HgO (s) and
not HgO (g) (e.g., Pal and Ariya, 2004;  Ariya et al., 2009), suggesting that pure gas phase
oxidation of elemental mercury by $O_3$ may not occur in the atmosphere. However, Calvert and
Lindberg (2005) proposed an alternative mechanism that would make this reaction potentially
viable in the atmosphere (Subir et al., 2011). The reaction may start with the formation of a
metastable $HgO_3$ (g) molecule which then decomposes to OHgOO (g) and thereafter
transforms to HgO (s) and $O_2$ (g).
$Hg(0)\,(g) + O_3\,(g) \rightarrow HgO\,(g) + O_2\,(g)$ (R1)
**3.6.2  Influence of heterogeneous surfaces**
As suggested by Subir et al. (2011), the influence of heterogeneous surfaces of water droplets,
snow, ice and aerosols should be taken into account when attempting to describe mercury
chemistry in the atmosphere. O'Concubhair et al. (2012) showed that freezing an acidic
solution containing nitrite or hydrogen peroxide can oxidize dissolved gaseous mercury in the
dark. Nitrous acid and hydrogen peroxide are present on the Antarctic plateau (Huey et al.,
2004;  Hutterli et al., 2004). As suggested by Dommergue et al. (2012), similar processes
could occur in the snow or on surface hoar at Concordia Station in winter. In 2013, the height
of measurement had a significant influence on the decline over time of Hg(0) concentrations
(ANCOVA test, $p$ value $< 2.10^{-16}$), with a steeper decrease at 25 cm than at 1070 cm. This
result suggests that snowpack may act as a sink for mercury, enhancing the deposition rate
due to heterogeneous reactions, through absorption of oxidation products, and/or physical
sorption/condensation of Hg(0) on surface snow.
In spite of the observed decreasing trend of Hg(0) concentrations in ambient air, total mercury
concentrations in surface snow samples did not significantly increase over time in winter (see
Fig. 4). Using a snow density of 300 kg/m$^3$ a loss of 0.30 ng/m$^3$ over a period of three months
in a mixing layer of 50 m high would lead to a 5.0 ng/L increase in the first cm of the
snowpack. Given the variability of chemical species deposition onto the snow surface, and the
occurrence of either fresh snowfall or blowing snow, this 5.0 ng/L increase over a period of
three months could not be detected in our weekly surface snow samples.
Despite the overall decreasing trend in winter, Hg(0) concentration exhibited abrupt increases
when moist and warm air masses from lower latitudes occasionally reached Concordia


Station. This is, for example, evidenced on 13 June 2012 by an increase of 0.25 ng/m$^3$ of the
Hg(0) concentration, an increase of temperature at 10 m a.g.l. from -63 to -26 °C, and a high
integrated water vapor column (see Fig. 12).

### 3.7    Implications at a continental scale

Depletion events of atmospheric Hg(0) that have been observed in the Artic and at various
coastal Antarctic stations have been associated with O$_3$ depletions, where Hg(0) and O$_3$
concentrations are positively correlated (Simpson et al., 2007). Increases in both Hg(II) and
Hg(p) have been reported in conjunction with decreases of Hg(0) (Lu et al., 2001; Lindberg et
al., 2002; Aspmo et al., 2005). Conversely, low Hg(0) concentrations that were not correlated
or anti-correlated with O$_3$ were observed at Neumayer and Troll (Temme et al., 2003;
Pfaffhuber et al., 2012), while elevated Hg(II) concentrations (up to 0.33 ng/m$^3$) were
recorded at Terra Nova Bay in the absence of Hg(0)/O$_3$ depletion (Sprovieri et al., 2002). The
continuous oxidation of Hg(0) in summer (see section 3.2) and multi-day Hg(0) depletion
events observed at Concordia Station in January/February (see section 3.5) are expected to
result in the build-up of an inland atmospheric reservoir enriched in Hg(II) and depleted in
Hg(0) in the summer. Due to strong katabatic winds flowing out from the Antarctic Plateau –
generated by the negative buoyant force that develops in the stable cooled layer along the ice
sheet slopes (Gallée and Pettré, 1998) –, a fraction of this inland atmospheric reservoir can be
transported toward the coastal margin. The influence of the flows from the Antarctic plateau
on coastal locations varies depending on the location. As demonstrated by Parish and
Bromwich (1987) and Parish and Bromwich (2007), the volume of air moving off inland
Antarctica toward the coastal margin displays significant spatial variability due to the
topographic slope and orientation of the underlying ice sheets. Northward transport of air
from the plateau is enhanced in a few locations called confluence zones – e.g., the broad
region upslope from the Ross Ice Shelf at 175°E and the area near Adélie Land at 142°E
(Parish and Bromwich, 1987, 2007) – but can be sporadically observed elsewhere explaining
observations at Neumayer, Troll, or Terra Nova Bay (Temme et al., 2003; Sprovieri et al.,
2002; Pfaffhuber et al., 2012). Monitoring atmospheric mercury at a coastal station situated
close to a confluence zone could provide new insights regarding the extent of the transport of
reactive air masses from the Antarctic plateau. This topic will be addressed in a companion
papier (Angot et al., this issue). The Antarctic continent shelters unconventional atmospheric
pathways of mercury reactivity both in winter and in summer. Its role should be taken into
account in the modeling of the global geochemical cycle of mercury.




## 4  Conclusion


Mean summertime atmospheric Hg(0) concentration was ~ 25% lower compared to values
recorded at other Antarctic stations at the same period of the year, suggesting a continuous
oxidation of atmospheric Hg(0) within the shallow boundary layer as a result of the high
oxidative capacity of the Antarctic plateau atmosphere at this period of the year. This
hypothesis is further supported by high total mercury concentrations in surface snow samples
measured at the station (up to 74 ng/L). Our results confirm short-term observations by
Brooks et al. (2008) and Dommergue et al. (2012) of chemical exchange at the air/snow
interface. During the sunlit period, Hg(0) concentration peaked in the upper layers of the
snowpack. Summertime Hg(0) concentration in ambient air exhibited a pronounced diurnal
cycle likely due to large emissions from the snowpack as a response to daytime heating. Our
observations also reveal a decrease of atmospheric mercury concentrations in winter (24-h
darkness) likely due to a gas-phase oxidation and/or heterogeneous reactions. Interestingly,
this decreasing trend has never been observed elsewhere. Finally, we reveal the occurrence of
multi-day to weeklong depletion events of Hg(0) in ambient air in summer, that are not
associated with depletion of $O_3$, and likely result from a stagnation of air masses on the
plateau triggering an accumulation of oxidants in the shallow boundary layer. This behaviour
is radically different from what is usually observed in the Arctic where only mercury
depletion events that were associated with $O_3$ depletion (and with a Hg(0)/$O_3$ correlation)
have been highlighted so far. According to observations at coastal Antarctic stations, the
reactivity observed at Concordia Station can be transported at a continental scale by strong
katabatic winds. Our understanding of the atmospheric mercury chemistry on the Antarctic
plateau is currently limited by the lack of continuous halogens measurements. Our findings
point out new directions for future kinetic, observational, and modeling studies.

## Acknowledgements


We thank A. Barbero and the rest of the overwintering crew: S. Aubin, C. Lenormant, and R.
Jacob. We also gratefully acknowledge M. Barret for the development of a QA/QC software
program, D. Liptzin for the calculation of the Obukhov length and friction velocity, M.
Legrand for the 2012 ozone data, C. Genthon for the meteorological data, E. Vignon for
helpful discussion, and B. Jourdain and X. Faïn for their help in the field. This work



contributed to the EU-FP7 project Global Mercury Observation System (GMOS –
www.gmos.eu) and has been supported by a grant from Labex OSUG@2020 (Investissements
d'avenir – ANR10 LABX56), and the Institut Universitaire de France. BQ and PR
acknowledge ANR CLIMSLIP and HAMSTRAD Program 910, respectively. Logistical and
financial support was provided by the French Polar Institute IPEV (Program 1028, GMOstral
and Program 1011, SUNITEDC), and a grant from the U.S. National Science Foundation
(NSF, PLR#1142145). Computing resources for FLEXPART simulations were provided by
the IPSL CICLAD/CLIMSERV mesocenter. Meteorological data were obtained thanks to
LEFE/IMAGO programs CLAPA and GABLS4, IPEV program CALVA/1013, and
Observatoire des Sciences de l'Univers de Grenoble (GLACIOCLIM observatory).



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



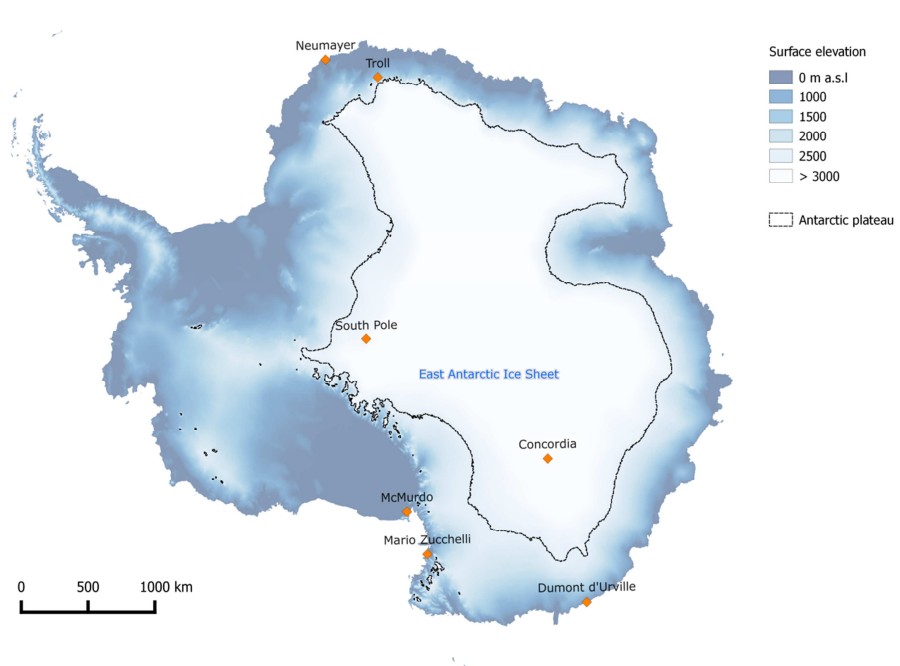

**Figure 1:** Map of Antarctica showing surface elevation (meters above sea level, m a.s.l) and the position of stations where atmospheric mercury measurements have been performed with modern online instruments. The black line shows the periphery of the high altitude plateau (> 2500 m a.s.l).





**a)**

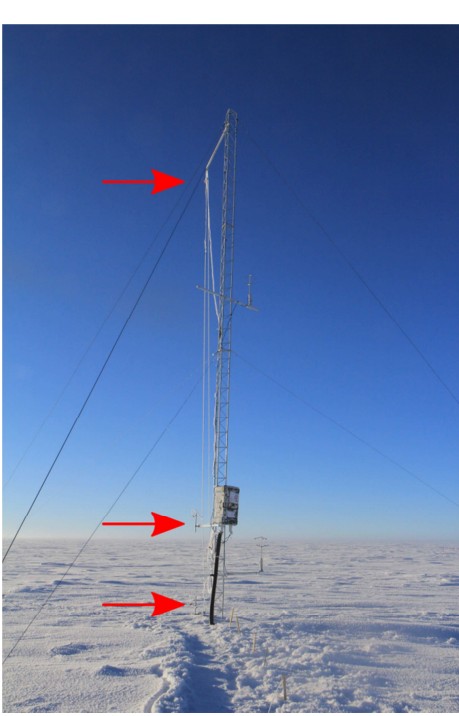

**b)**

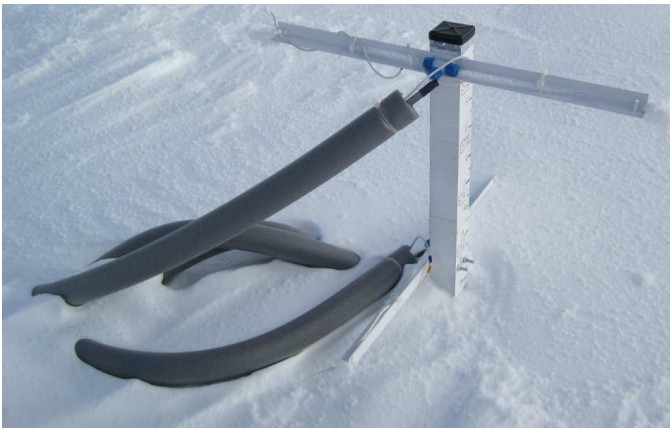

**Figure 2:** Photographs showing **a)** the meteorological tower with the three gas inlets (red arrows) at 1070 cm, 210 cm and 25 cm above the snow surface (photo credit: B. Jourdain), and **b)** one of the snow towers with the two sampling inlets above the snowpack at 50 and 10 cm (photo credit: D. Helmig) .





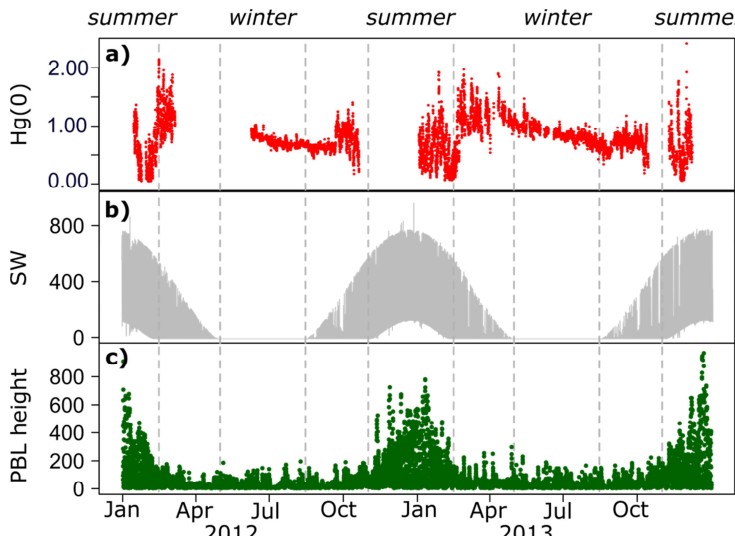

**Figure 3:** Annual variation in 2012 and 2013 of **a)** hourly-averaged Hg(0) concentrations (in ng/m³) at 500 cm and 25 cm above the snow surface in 2012 and 2013, respectively, **b)** downwelling shortwave (SW) radiation (in W/m²), and **c)** planetary boundary layer (PBL) height (in m). The vertical dashed lines represent seasonal boundaries.





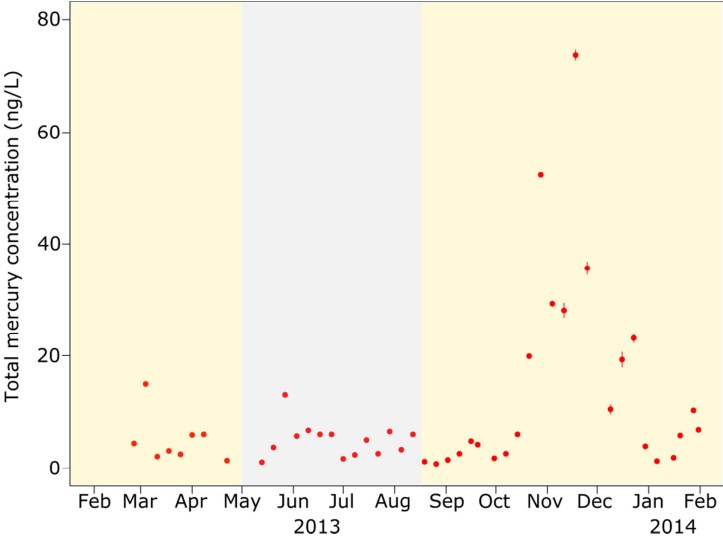

**Figure 4:** Total mercury concentration (ng/L), along with standard errors, in surface snow samples collected weekly at Concordia Station from February 2013 to January 2014. Dark period (winter) highlighted in grey, sunlit period highlighted in yellow. Total mercury concentrations were elevated (up to 74 ng/L) in November-December 2013 (summer).




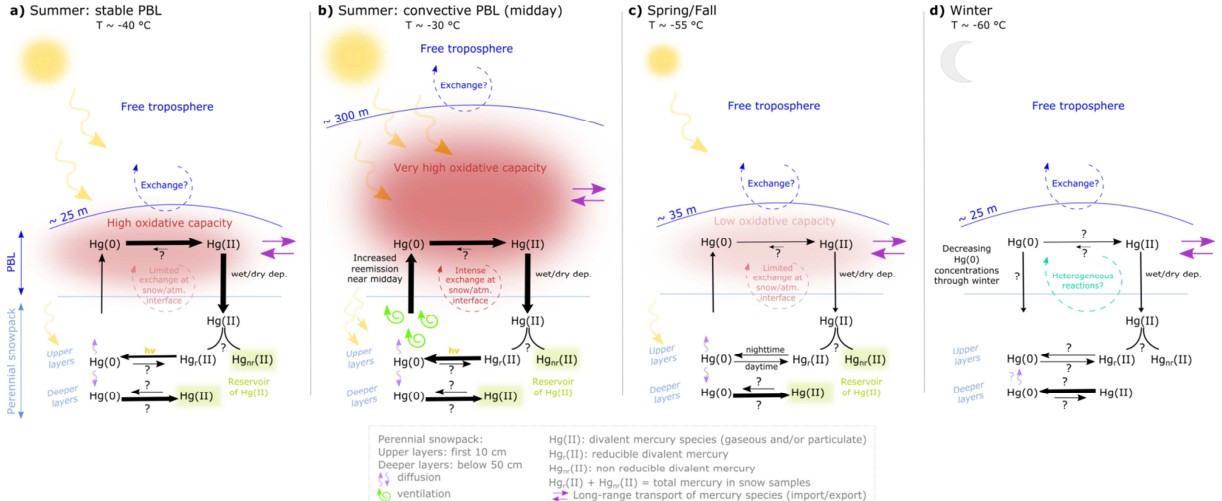

**Figure 5:** Schematic diagram illustrating the processes that govern the Hg(0) budget at Concordia Station **a)** in summer under stable Planetary Boundary Layer (PBL) conditions, **b)** in summer under convective PBL conditions, **c)** in spring/fall, and **d)** in winter. In summer, Hg(0) is continuously oxidized due to the high oxidative capacity of the boundary layer and a large amount of divalent mercury species deposit onto the snowpack. A fraction of deposited mercury can be reduced (the reducible pool, Hg$_r$(II)) in the upper layers of the snowpack and subsequently reemitted to the atmosphere as Hg(0). Hg(0) emission from the snowpack maximizes near midday likely due to increased ventilation as a response to daytime heating. Oxidation of Hg(0) dominates in the deeper layers of the snowpack leading to the formation of a Hg(II) reservoir. In spring/fall, the balance of reduction and oxidation processes within the upper layers of the snowpack differs from summertime: oxidation dominates during the day, reduction at night. In winter, Hg(0) is produced in the deeper layers of the snowpack likely as a result of the reduction of Hg(II) species accumulated during the sunlit period. Ambient Hg(0) concentrations exhibit a 20 to 30% decrease through winter.





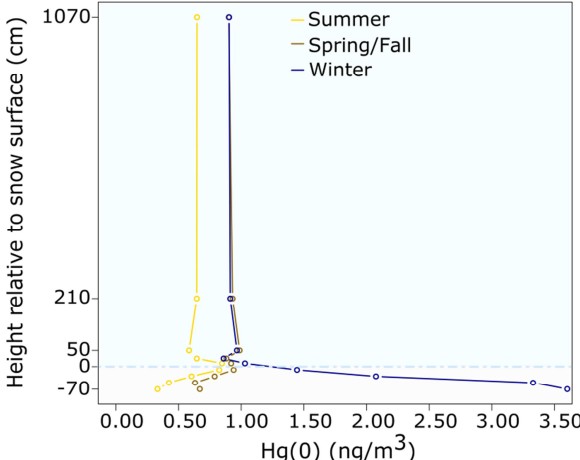

**Figure 6:** Mean Hg(0) concentration (ng/m³) measured at various heights above and below the snow surface (cm) at Concordia Station in summer (yellow), spring/fall (brown), and winter (dark blue) The horizontal light blue dashed line represents the snow surface.





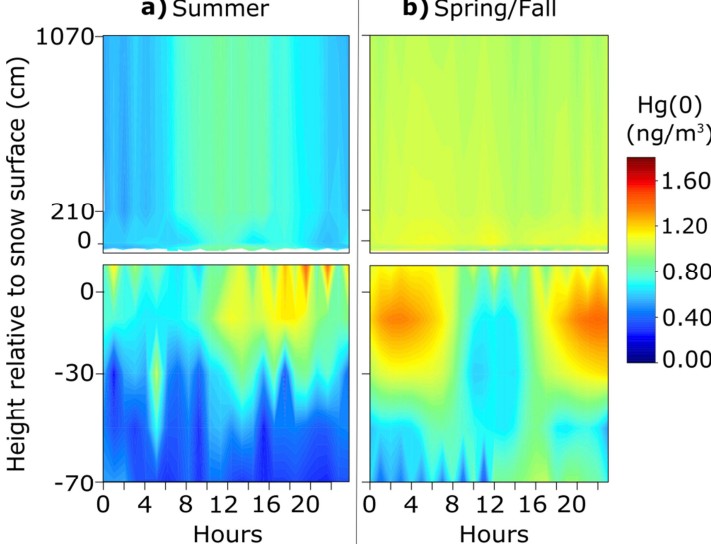

**Figure 7:** Hourly (local time) mean atmospheric and interstitial air Hg(0) concentrations in **a)** summer, and **b)** spring/fall. The vertical axis is the height of measurement relative to the snow surface (in cm). Color contours show Hg(0) concentrations (in ng/m$^3$).





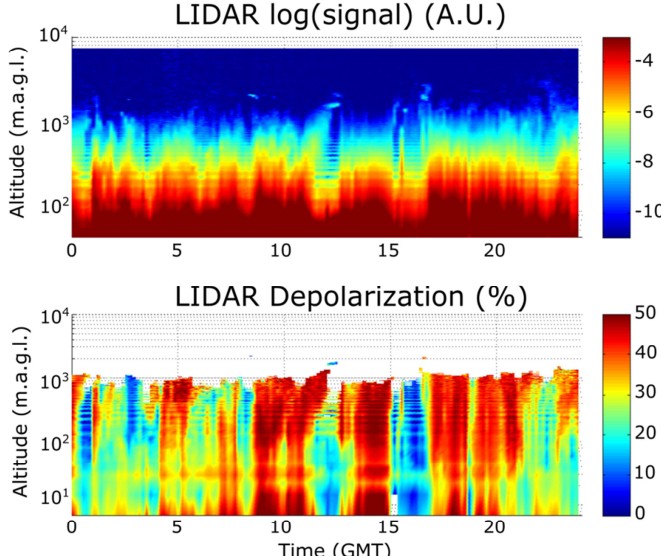

**Figure 8:** Lidar backscatter (upper panel) and depolarization ratio (lower panel) measured at Concordia Station on 24 February 2013. Ice precipitation was observed in the first 1000 m of the atmosphere, with only two small liquid-water clouds around 8 and 12 UTC.





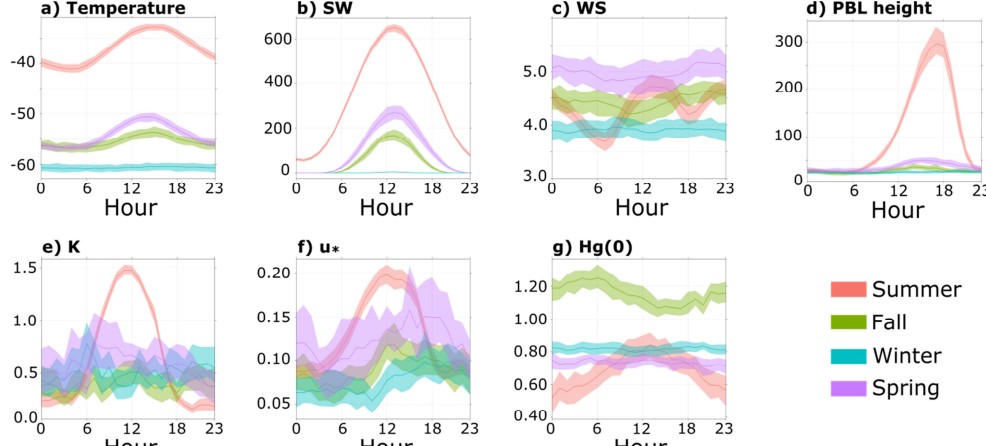

**Figure 9:** Hourly (local time) mean variation, along with the 95% confidence interval for the mean, of: **a)** temperature (in °C) at 3 m above the snow surface, **b)** downwelling shortwave (SW) radiation (in W/m$^2$) according to the MAR model simulations, **c)** wind speed at 3 m above the snow surface (in m/s), **d)** planetary boundary layer (PBL) height (in m) according to the MAR model simulations, **e)** Eddy diffusivity ($K$, in m$^2$/s), **f)** friction velocity ($u_*$, in m/s), and **g)** Hg(0) concentration (in ng/m$^3$), in summer (red), fall (green), winter (blue), and spring (purple).





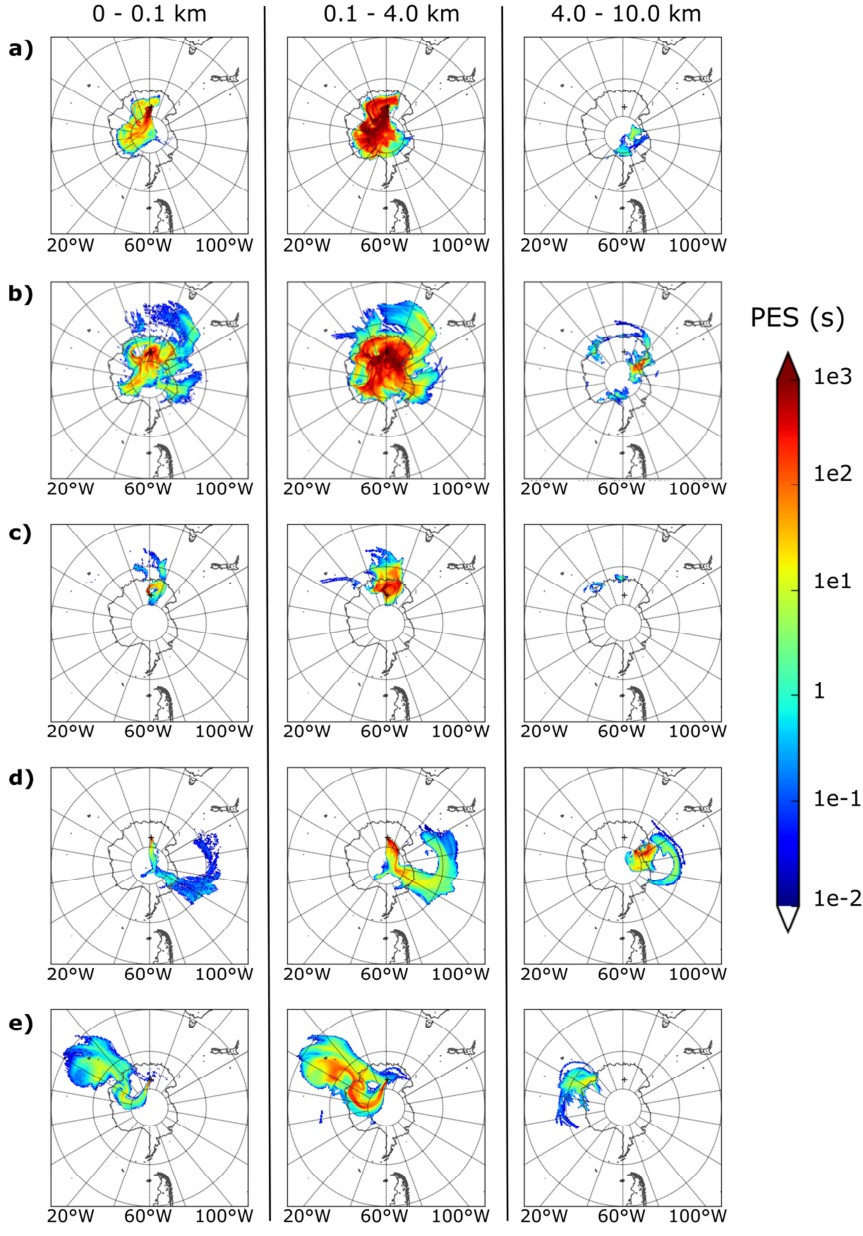

**Figure 10:** Back trajectories for the 3 layers of altitude colored according to the potential emission sensitivity (PES, in seconds) **a)** from 19 January to 8 February 2012, **b)** from 5 to 20 February 2013, **c)** on 10 February 2012, **d)** on 22 February 2013, and **e)** on 11 September 2013. Note that PES in a particular grid cell is proportional to the particle residence time in that cell.





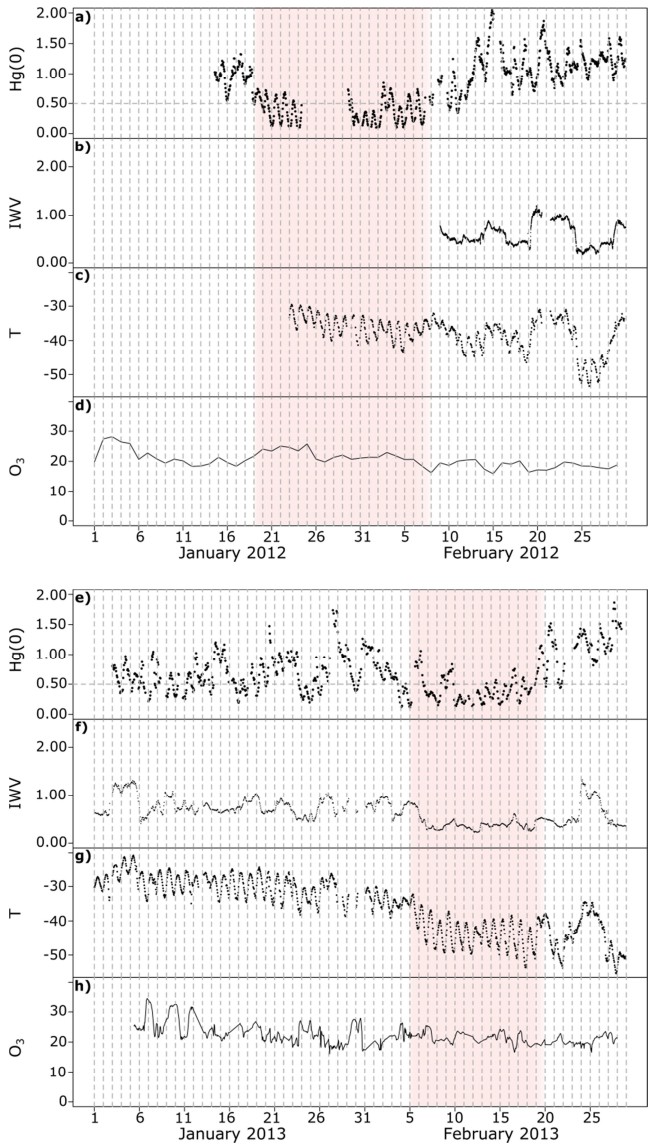

**Figure 11:** Top: January and February 2012 cycle of: **a)** hourly-averaged Hg(0) concentrations (in ng/m$^3$), **b)** Integrated Water Vapor (IWV, kg/m$^2$), **c)** Temperature (in °C) at 10 m above ground level, and **d)** ozone (O$_3$, daily mean) mixing ratios (ppbv). Hg(0) was low from 19 January to 8 February (period highlighted in red) while O$_3$ showed no abnormal variability. Bottom: January and February 2013 cycle of: **e)** hourly-averaged Hg(0) concentrations (in ng/m$^3$), **f)** Integrated Water Vapor (IWV, kg/m$^2$), **g)** Temperature (in °C) at 10 m above ground level, and **h)** ozone (O$_3$) mixing ratio (ppbv). Hg(0), IWV, and temperature were low from 5 to 20 February (period highlighted in red) while O$_3$ showed no abnormal variability.





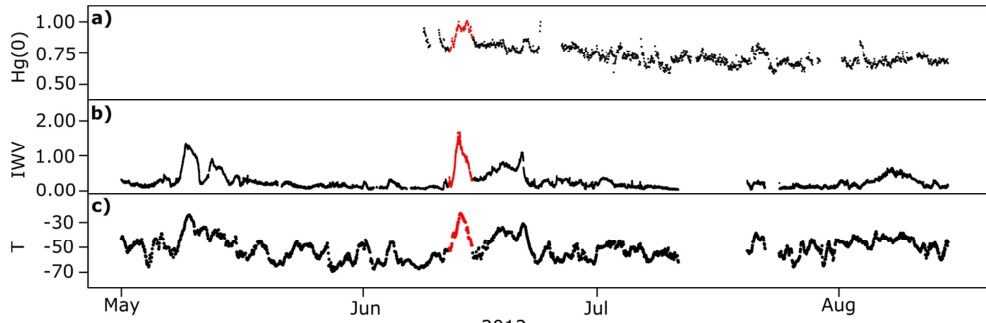

**Figure 12:** Year 2012 wintertime record of: **a)** hourly-averaged Hg(0) concentrations (in ng/m³), **b)** Integrated Water Vapor (IWV, kg/m²), and **c)** Temperature (T, °C) at 10 m above ground level. Hg(0), temperature, and IWV increased from June 12 to 15 (in red) suggesting transport of moist and warm air masses originating from lower latitudes