# Peer review of "New insights into the atmospheric mercury cycling in Central Antarctica and implications at a continental scale"

_Atmospheric Chemistry and Physics, 2016_

## Referee Comment (RC1) · Anonymous Referee #1 · 8 Apr 2016

General comments The manuscript is well written and presents a novel dataset of atmospheric mercury and mercury in snow interstitial air covering slightly more than one annual cycle. Mercury measurements from Antarctica is generally scarce in particular good data covering more than the summer season, so this manuscript will definitely be a longed for addition to the pool of mercury data for the scientific community, both for experimentalists and modelers. The references used are recent and relevant. The structure of the result and discussion chapter is complicated. The text jumps between different environmental compartments (atmosphere and snow interstitial air) and seasons, which makes it ponderous to follow the discussion. The authors should consider re-arranging the sections within the results and discussion chapter. Scientific com-

ments Line 65-70: The manuscript text says that the Antarctic plateau was first considered to be chemically- inactive for atmospheric species including Hg and a paper from 2008 is cited. The manuscript text further says that it turned out to be highly active, now citing a paper from 2001 and 2007. To me there is a lack of logic in this argument. Line 131: Was mercury saturated manually? I assume you mean manually injecting saturated mercury. Line 144-146: QA/QC; Interesting that you use internal standard on the 2600, what was used as internal standard? Why not use commercially available control samples? Line 202-204: This was an elegant way of defining seasons! Line 211-212: Troll can hardly be called a coastal site as it is situated almost 250 km from the coast and at almost 1500 masl. However, due to its location Troll experiences air from both the Antarctic plateau and the southern Ocean, but it is not a coastal site. Line 259-263: I like the figure showing the vertical distribution of Hg(0) concentrations, however I think it could also be interesting for the audience to present a figure showing time series of Hg(0) in SIA as well. The atmospheric Hg(0) time series seems well covered in other figures. Line 356 and throughout the paragraph: I am not quite sure but I believe it is general consensus regarding the use of the term "depletion event" being oxidation of Hg resulting from bromine photochemistry and high correlation with tropospheric ozone depletions. The multi-day low concentrations described in chapter 3.5 are a different mechanism and should consequently be called something else. Line 391 and throughout chapter 3.6: A mechanism for reduced Hg(0) during winter is proposed, and it is also mentioned that this reductions are not observed at Troll or Neumayer. Any thought on why this reaction mechanism do not occur at Troll or Neymayer? Do you believe this reaction mechanism occur throughout the Antarctic plateau? Additionally, as you have O3 measurements it would be interesting to have them presented in fig 12 alongside Hg(0) since O3 is suggested to be involved in the Hg(0) wintertime decrease. The authors should perhaps also have a look at a very recent paper by Nerentorp Mastromonaco et al., 2016 (Atmospheric Environment) where winter depletions are present and to check whether this may have any relevance for the winter decreasing trend observed in this study.

Technical comments: Line 412: Several characters in the equation disappear in print Figure 6: It is difficult to tell the difference between summer and spring/fall colours in the figure and I would very much like to see something similar to box and whisker plots to be able to tell the concentration distribution at each height at each season. Figure 9: Hg(0) concentrations, which measurement height do the results represent? Figure 11: Same comment as above, please indicate at which height the Hg(0) measurements are from. Figure 12: same comment as above.
* * *

---

## Referee Comment (RC2) · Anonymous Referee #2 · 31 May 2016

**Review comments for "New insights into the atmospheric mercury cycling in Central Antarctica and implications at a continental scale" by Angot et al., acp-2016-144**

**General comments:**

This paper describes year-round measurements of Hg(0) in the atmosphere and snow on the Antarctic plateau along with ancillary measurements. These comprise a novel data set that is very valuable for understanding the global atmospheric (and cryospheric) mercury cycle. Given the value of these data and the difficulty in duplicating them, I would encourage the authors to make the complete data set available in some capacity (e.g. as a supplementary file, or a link to a data repository) in order to aid modellers, etc., in using these measurements to advance further research.

The analysis and interpretation is largely sound, with a few gaps as identified in the comments below. I do agree with the first reviewer that the organization of the Results and Discussion could be improved. I recommend the publication of this paper after the minor issues discussed below have been addressed.

**Specific comments:**

l. 38-39: "according to observations at coastal Antarctic stations" is vague; this is used elsewhere in the paper (l. 488) and is not very enlightening. Can you summarize the evidence you are using to draw this conclusion? Perhaps in Section 3.7? There are references there but the observations are not described.

l. 53: "rapid deposition" is relative. You later describe a reservoir of gaseous Hg(II), which can hardly be expected if the deposition lifetime is very rapid.

Section 2.3: What is the estimated precision of your Hg(0) measurements?

Section 3.1: (a) This is titled "seasonal variation" but mostly summarizes annual values and spatial/vertical differences. (b) Can you identify what the ± values are? Standard deviation? Confidence limits on the mean? (c) What statistical test was used to determine that your values were lower than the Troll and Neumayer – Mann-Whitney as well?

How did the 25 cm inlet met tower values compare with the 50 cm and 10 cm snow tower inlet values overall? Fig. 6 suggests there was some offset between the met tower and snow tower inlets, at least in winter and possible spring/fall. Is the sampling coverage the same? Could there be an effect of heated/non-heated lines, or the length of the sampling lines?

Fig. 3: Why did you choose the 25 cm inlet to show? Can you add a time series or two (shallow/deep) for the snowpack data?

Fig. 4: I only see error bars on a few points. Are these the only ones with replicates? How many replicates were done in those cases? A line or two in the caption to explain this would be helpful.

Section 3.2: You refer to "continuous" oxidation in the summer. What do you mean by that? It is clear there is net oxidation, but I am not sure you have shown it is continuous.

Fig. 7: This figure is a nice attempt to visualize the data, but it is rather confusing. Does the shading represent some sort of unspecified interpolation? Do the top boxes represent the met tower and the bottom the snow tower? In that case, why do the top boxes extend down below zero and the bottom ones not go up to 50 cm? If not, why don't the top and bottom agree within the overlap region?

Fig. 8: I think this figure is not crucial to the paper, since it is only being used to support a suggested mechanism for a single extreme value. I think you can make that suggestion without an additional figure, though it is up to your discretion.

Section 3.3.2: You mention a shift from oxidation to reduction at the beginning of winter, but it would be very helpful to see the time series of Hg(0) at depth (as mentioned above) – is it a sudden drop to a stable "winter" value, or is there a longer trend over the winter to accompany the atmospheric decline?

Section 3.4.1: Your summary is a bit confusing (ll. 339-343). I think you are saying by "continuous" that (i) has a week diurnal cycle and by "important" that (iii) has a strong diurnal cycle, resulting in the observed concentration pattern. Can you say this more clearly? I'm not sure what "important" means in (ii). Clearly it is important to the surface snow THg, but I'm not sure how this is related to the Hg(0) diurnal cycle.

Section 3.4.2: Where is this Hg(0) building up from? Presumably the snow, but it's not mentioned. Why is the fall concentration higher than spring? The reservoir of Hg in the summer snow?

Fig. 9g: What inlet is the Hg(0) cycle from?

Section 3.5: It's a bit odd to refer to Fig. 10e first. I suggest you rearrange the figure to make this 10a.

Section 3.6: (a) Given you have a single winter of data, and the decline is not seen at the other stations, can you eliminate instrument drift (e.g. trap poisoning)? Were the external calibrations before and after the winter consistent? (b) Why do you not include dry deposition of Hg(0) as a possible mechanism for this decrease? Given the low BL, what deposition flux would be needed to remove the observed amount of Hg(0)? How does this compare to other observations/calculations (Cobbett et al 2007, Zhang et al 2009)? I think it's quite similar. It would also account for the gradient in the decrease (3.6.2). (c) Speaking of which, you don't report in 3.1 if there are any seasonal differences in the three met tower inlets. Your discussion of the winter data suggests there would be. (d) This section is poorly organized. I suggest removing the sub-sections since you basically discount the gas-phase reaction without doing so explicitly. (e) l. 402: Why don't you report your O3 data instead of (or as well as) referring to another paper? Also, are there O3 data at Troll or Neumayer that suggest that a winter reaction with O3 would not also happen there?

Section 3.7: A bit more detail about the observations that are attributed to transport from the plateau (ll. 457-462) would be helpful, as mentioned above.

Section 4: (a) I'd like to see a mention of the intriguing winter subsurface Hg(0) peak in here. (b) Maybe change "heating" to "snowpack ventilation" or "ventilation and heating" in l. 479. (c) In l. 481 I would change "likely" to "possibly".... And do you really think gas-phase oxidation is even that likely? Your

earlier discussion suggests not. You may need to add dry deposition of Hg(0) as well, depending what you find.

**Technical corrections:**

l. 32: change "never been observed" to "not been reported"

l. 46: change "contamination" to "contaminant"

l. 246: change "bound" to "bind"

l. 315: change "significant and daily" to "significant daily"

l. 316: change "all along" to "throughout"

l. 358: suggest changing "explosions" to "so-called 'bromine explosions'" to avoid leaving unfamiliar readers with the impression there are actual explosions.

l. 378: change "these depletions of Hg(0)" to "the depletions of Hg(0) reported here"

---

## Author Response (AR1)

**"New insights into the atmospheric mercury cycling in Central Antarctica and implications at a continental scale" by H. Angot et al.**

**Response to referee comments by Referee #1.**

We thank this anonymous referee for insightful questions and comments. We provide below a point-by-point reply to the comments (points raised by the referee in bold, changes made in the manuscript in red).

1. General comments

**The manuscript is well written and presents a novel dataset of atmospheric mercury and mercury in snow interstitial air covering slightly more than one annual cycle. Mercury measurements from Antarctica is generally scarce in particular good data covering more than the summer season, so this manuscript will definitely be a longed for addition to the pool of mercury data for the scientific community, both for experimentalists and modelers. The references used are recent and relevant. The structure of the result and discussion chapter is complicated. The text jumps between different environmental compartments (atmosphere and snow interstitial air) and seasons, which makes it ponderous to follow the discussion. The authors should consider re-arranging the sections within the results and discussion chapter.**

We agree with the referee regarding the structure of the results and discussion chapter. In the revised manuscript we have changed the structure as follows in order to avoid jumps between different environmental compartments (atmosphere and snow interstitial air) and seasons.

3. Results and Discussion

3.1 Hg(0) concentrations in ambient air

      3.1.1 Spring
      3.1.2 Summertime
              a) Oxidation of Hg(0) in ambient air and Hg(II) deposition onto snowpack
              b) Multi-day depletion events of atmospheric Hg(0)
              c) Hg(0) diurnal cycle

      3.1.3 Fall
      3.1.4 Winter

3.2 Hg(0)/Hg(II) redox conversions within the snowpack

      3.2.1 Sunlit period
      3.2.2 Winter

4. Implications at a continental scale

2. Scientific comments

**- Line 65-70: The manuscript text says that the Antarctic plateau was first considered to be chemically inactive for atmospheric species including Hg and a paper from 2008 is cited. The manuscript test further says that it turned out to be highly active, now citing a paper from 2001 and 2007. To me there is a lack of logic in this argument.**

We agree. In the revised manuscript we now refer to an older reference:

"The Antarctic plateau (...) was first considered to be chemically-inactive and a giant cold trap for atmospheric species, including mercury (e.g., Lambert et al., 1990). It turned out to be highly photochemically active (Davis et al., 2001; Grannas et al., 2007)…"

**- Line 131: Was mercury saturated manually? I assume you mean manually injecting saturated mercury.**

Yes indeed. This has been corrected in the revised manuscript.

**- Line 144-146: QA/QC; Interesting that you use internal standard on the 2600, what was used as internal standard? Why not use commercially available control samples?**

These questions have been addressed in the revised manuscript:

"The instrument was calibrated with the NIST SRM-3133 mercury standard. Quality assurance and quality control included the analysis of analytical blanks, replicates, and internal standards (Reference Waters for mercury: HG102-2 at 22 ng/L from Environment Canada)."

**- Line 202-204: This was an elegant way of defining seasons!**

Thank you.

**- Line 211-212: Troll can hardly be called a coastal site as it is situated almost 250 km from the coast and at almost 1500 masl. However, due to its location Troll experiences air from both the Antarctic plateau and the southern Ocean, but it is not a coastal site.**

We agree that Troll can hardly be called a coastal site. This has been corrected in the revised manuscript:
"(…) Hg(0) concentrations are lower than annual averages reported at near-coastal or coastal Antarctic stations..."

**- Line 259-263: I like the figure showing the vertical distribution of Hg(0) concentrations, however I think it could also be interesting for the audience to present a figure showing time series of Hg(0) in SIA as well. The atmospheric Hg(0) time series seems well covered in other figures.**

A Figure displaying the annual variation of Hg(0) concentrations in the snow interstitial air collected at the various inlets of the two snow towers has been added in the revised manuscript:

[Figure]

"Figure 10: Annual variation of hourly-averaged Hg(0) concentrations (in ng/m³) in the snow interstitial air collected at the various inlets of the two snow towers: **a)** snow tower #1, **b)** snow tower #2. Note that we regularly experienced technical problems on snow tower #2 leading to missing values."

**- Line 356 and throughout the paragraph: I am not quite sure but I believe it is general consensus regarding the use of the term "depletion event" being oxidation of Hg resulting from bromine photochemistry and high correlation with tropospheric ozone depletions. The multi-day low concentrations described in chapter 3.5 are a different mechanism and should consequently be called something else.**

We are not sure that renaming a depletion of mercury specific to this site is warranted. A "depletion event" is just a depletion of Hg(0), and is not specific to the halogen chemistry. We do understand that some may associate it but "depletion event" is a more generic term than that.

**- Line 391 and throughout chapter 3.6: A mechanism for reduced Hg(0) during winter is proposed, and it is also mentioned that this reductions are not observed at Troll or Neumayer. Any thought on why this reaction mechanism does not occur at Troll or Neymayer? Do you believe this reaction mechanism occur throughout the Antarctic plateau?**

The reason why this reaction mechanism does not occur at Troll or Neumayer is unclear. This could be due to meteorological conditions on the Antarctic plateau (e.g., temperature, relative humidity, boundary layer dynamics). Further research is clearly needed. Our sense is that this reaction mechanism might occur throughout the Antarctic plateau, but the spatial distribution of Hg(0) measurements should be improved.

**- Additionally, as you have O3 measurements it would be interesting to have them presented in fig 12 alongside Hg(0) since O3 is suggested to be involved in the Hg(0) wintertime decrease.**

Figure 12 shows that – despite the overall decreasing trend in winter – Hg(0) concentration exhibited abrupt increases when moist and warm air masses from lower latitudes occasionally reached Concordia Station. Key parameters are therefore Hg(0), integrated water vapor, and temperature. We do not think that adding $O_3$ measurements here would be relevant. However, the annual variation of $O_3$ measurements in 2012 and 2013 has been added in Figure 3 of the revised manuscript.

[Figure]

"Figure 3: Annual variation in 2012 and 2013 of a) hourly-averaged Hg(0) concentrations (in ng/m$^3$) at 500 cm and 25 cm above the snow surface in 2012 and 2013, respectively, b) downwelling shortwave (SW) radiation (in W/m$^2$), c) planetary boundary layer (PBL) height (in m), and d) ozone (O$_3$, daily mean in 2012 and hourly mean in 2013) mixing ratios (in ppbv). The vertical dashed lines represent seasonal boundaries."

**- The authors should perhaps also have a look at a very recent paper by Nerentorp Mastromonaco et al., 2016 (Atmospheric Environment) where winter depletions are present and to check whether this may have any relevance for the winter decreasing trend observed in this study.**

Thank you for the suggestion. Nerentorp Mastromonaco et al. (2016) observed Atmospheric Mercury Depletion Events (AMDEs) during Antarctic winter over sea ice areas and proposed a dark mechanism in the marine boundary layer. It should be noted that such events have never been reported on the Antarctic continent (coastal or inland stations). Given the distance of Concordia station from sea ice areas a similar dark mechanism seems unlikely.

**3. Technical comments**

**- Line 412: Several characters in the equation disappear in print**

Thanks for noticing that. We will make sure there is no such problem in the final version of the manuscript.

**- Figure 6: It is difficult to tell the difference between summer and spring/fall colours in the figure and I would very much like to see something similar to box and whisker plots to be able to tell the concentration distribution at each height at each season.**

The color for spring/fall values has been changed in the revised manuscript. It should now be easier to tell the difference between summer and spring/fall colors. The concentration distribution at each height at each season can be inferred from the Figure we have added in the revised manuscript (see comment line 259-263).

**- Figure 9: Hg(0) concentrations, which measurement height do the results represent?**

This has been added in the caption of the revised manuscript:

"Figure 7: Hourly (local time) mean variation, along with the 95% confidence interval for the mean, of: **a)** Hg(0) concentration (in ng/m$^3$) at 25 cm above the snow surface, **b)** downwelling shortwave (SW) radiation (in W/m$^2$) according to the MAR model simulations, **c)** temperature (in °C) at 3 m above the snow surface, **d)** wind speed at 3 m above the snow surface (in m/s), **e)** planetary boundary layer (PBL) height (in m) according to the MAR model simulations, **f)** friction velocity ($u_*$, in m/s), and **g)** Eddy diffusivity ($K$, in m$^2$/s) in summer (red), fall (green), winter (blue), and spring (purple). Note that the hourly mean variation of Hg(0) concentration in summer is similar at the three inlets of the meteorological tower".

**- Figure 11: Same comment as above, please indicate at which height the Hg(0) measurements are from.**

This has been added in the caption of the revised manuscript:

"Figure 6: Top: January and February 2012 cycle of: **a)** hourly-averaged Hg(0) concentrations (in ng/m$^3$) at 500 cm above the snow surface, **b)** Integrated Water Vapor (IWV, kg/m$^2$), **c)** Temperature (in °C) at 10 m above ground level, and **d)** ozone (O$_3$, daily mean) mixing ratios (ppbv). Hg(0) was low from 19 January to 8 February (period highlighted in red) while O$_3$ showed no abnormal variability. Bottom: January and February 2013 cycle of: **e)** hourly-averaged Hg(0) concentrations (in ng/m$^3$) at 210 cm above the snow surface, **f)** Integrated Water Vapor (IWV, kg/m$^2$), **g)** Temperature (in °C) at 10 m above ground level, and **h)** ozone (O$_3$) mixing ratio (ppbv). Hg(0), IWV, and temperature were low from 5 to 20 February (period highlighted in red) while O$_3$ showed no abnormal variability. Note that Hg(0) concentrations exhibited the same pattern at the three inlets of the meteorological tower from 5 to 20 February 2013".

**- Figure 12: same comment as above.**

This has been added in the caption of the revised manuscript:

"Figure 9: Year 2012 wintertime record of: **a)** hourly-averaged Hg(0) concentrations (in ng/m$^3$) at 500 cm above the snow surface, **b)** Integrated Water Vapor (IWV, kg/m$^2$), and **c)** Temperature (T, °C) at 10 m above ground level. Hg(0), temperature, and IWV increased from June 12 to 15 (in red) suggesting transport of moist and warm air masses originating from lower latitudes."

**4. References**

Davis, D., Nowak, J. B., Chen, G., Buhr, M., Arimoto, R., Hogan, A., Eisele, F., Mauldin, L., Tanner, D., Shetter, R., Lefer, B., and McMurry, P.: Unexpected high levels of NO observed at South Pole, Geophysical research letters, 28, 3625-3628, 2001.

Grannas, A. M., Jones, A. E., Dibb, J., Ammann, M., Anastasio, C., Beine, H. J., Bergin, M., Bottenheim, J., Boxe, C. S., Carver, G., Chen, G., Crawford, J. H., Domine, F., Frey, M. M., Guzman, M. I., Heard, D. E., Helmig, D., Hoffmann, M. R., Honrath, R. E., Huey, L. G., Hutterli, M., Jacobi, H.-W., Klan, P., Lefer, B., McConnell, J. R., Plane, J. M. C., Sander, R., Savarino, J., Shepson, P. B., Simpson, W. R., Sodeau, J., Von Glasow, R., Weller, R., Wolff, E. W., and Zhu, T.: An overview of snow photochemistry: evidence, mechanisms and impacts, Atmospheric Chemistry and Physics, 7, 4329-4373, 2007.

Lambert, G., Ardouin, B., and Sanak, J.: Atmospheric transport of trace elements toward Antarctica, Tellus B, 42, 76-82, 10.1034/j.1600-0889.1990.00009.x, 1990.

Nerentorp Mastromonaco, M., Gårdfeldt, K., Jourdain, B., Abrahamsson, K., Granfors, A., Ahnoff, M., Dommergue, A., Méjean, G., and Jacobi, H.-W.: Antarctic winter mercury and ozone depletion events over sea ice, Atmospheric Environment, 129, 125-132, 2016.

**"New insights into the atmospheric mercury cycling in Central Antarctica and implications at a continental scale" by H. Angot et al.**

**Response to referee comments by Referee #2.**

We would like to thank the anonymous referee for its time and useful comments towards the improvement of our manuscript. We provide below a point-by-point reply to the comments (points raised by the referee in bold, changes made in the manuscript in red).

1. General comments

**This paper describes year-round measurements of Hg(0) in the atmosphere and snow on the Antarctic plateau along with ancillary measurements. These comprise a novel data set that is very valuable for understanding the global atmospheric (and cryospheric) mercury cycle. Given the value of these data and the difficulty in duplicating them, I would encourage the authors to make the complete data set available in some capacity (e.g. as a supplementary file, or a link to a data repository) in order to aid modellers, etc., in using these measurements to advance further research.**

Mercury data reported in this paper are available upon request at http://sdi.iia.cnr.it/geoint/publicpage/GMOS/gmos_historical.zul. This has been added in the acknowledgments of the revised manuscript.

**The analysis and interpretation is largely sound, with a few gaps as identified in the comments below. I do agree with the first reviewer that the organization of the Results and Discussion could be improved. I recommend the publication of this paper after the minor issues discussed below have been addressed.**

We agree with the referee regarding the structure of the results and discussion chapter. In the revised manuscript we have changed the structure as follows in order to avoid jumps between different environmental compartments (atmosphere and snow interstitial air) and seasons.

3. Results and Discussion

3.1 Hg(0) concentrations in ambient air

     3.1.1 Spring
     3.1.2 Summertime
         a) Oxidation of Hg(0) in ambient air and Hg(II) deposition onto snowpack
         b) Multi-day depletion events of atmospheric Hg(0)
         c) Hg(0) diurnal cycle

     3.1.3 Fall
     3.1.4 Winter

3.2 Hg(0)/Hg(II) redox conversions within the snowpack

**2.  Specific comments**

**- l. 38-39: "according to observations at coastal Antarctic stations" is vague; this is used elsewhere in the paper (l. 488) and is not very enlightening. Can you summarize the evidence you are using to draw this conclusion? Perhaps in Section 3.7? There are references there but the observations are not described.**

"According to observations at coastal Antarctic stations" refers to the following sentence in Section 3.7 (Section *4* in the revised version of the manuscript, see previous comment regarding the structure of the Results and Discussion chapter):

"Conversely, low Hg(0) concentrations that were not correlated or anti-correlated with $O_3$ were observed at Neumayer and Troll (Temme et al., 2003;  Pfaffhuber et al., 2012), while elevated Hg(II) concentrations (up to 0.33 ng/m$^3$) were recorded at Terra Nova Bay in the absence of Hg(0)/$O_3$ depletion (Sprovieri et al., 2002)".

In an attempt to clarify this point, we have added (1) a reference to this sentence in lines 482-484 of the revised manuscript and (2) a reference to this Section in the conclusion:

(1) "but can be sporadically observed elsewhere explaining the aforementioned observations at Neumayer, Troll, or Terra Nova Bay ".

(2) "According to observations at coastal Antarctic stations (see section 4), the reactivity observed at Concordia Station can be transported at a continental scale by strong katabatic winds".

**- l. 53: "rapid deposition" is relative. You later describe a reservoir of gaseous Hg(II), which can hardly be expected if the deposition lifetime is very rapid.**

We agree. This has been corrected in the revised manuscript: "leading to the formation and subsequent  deposition"

**- Section 2.3: What is the estimated precision of your Hg(0) measurements?**

This has been specified in the revised version of the manuscript: "Based on experimental evidence, the average systematic uncertainty for Hg(0) measurements is of ~ 10 % (Slemr et al., 2015)".

**- Section 3.1: (a) This is titled "seasonal variation" but mostly summarizes annual values and spatial/vertical differences.**

This title has been removed in the revised manuscript (see previous comment regarding the structure of the Results and Discussion chapter).

**(b) Can you identify what the ± values are? Standard deviation? Confidence limits on the mean?**

± values refer to standard deviations. This has been clarified in the revised manuscript: "In summer, the mean atmospheric Hg(0) concentration was 0.69 ± 0.35 ng/m$^3$ (mean ± standard deviation)."

**(c) What statistical test was used to determine that your values were lower than the Troll and Neumayer – Mann-Whitney as well?**

None. This would require the entire distribution of Hg(0) concentrations at Troll and Neumayer. A two-sample z-test for comparing two means could be used. However, this test assumes that the two populations are normally distributed.

**- How did the 25 cm inlet met tower values compare with the 50 cm and 10 cm snow tower inlet values overall? Fig. 6 suggests there was some offset between the met tower and snow tower inlets, at least in winter and possible spring/fall. Is the sampling coverage the same? Could there be an effect of heated/non-heated lines, or the length of the sampling lines?**

Overall, the 25 cm inlet met tower values are lower than the 50 and the 10 cm snow tower inlet values. Values at the 50 and 10 cm snow tower inlets might be biased high due to contamination from the deeper inlets. Indeed, as mentioned in section 2.2 of the manuscript, sampling lines were purged continuously at 5 L/min on the met tower but intermittently at ~ 2-3 L/min on the two snow towers.

**- Fig. 3: Why did you choose the 25 cm inlet to show? Can you add a time series or two (shallow/deep) for the snowpack data?**

The decision to show Hg(0) at the 25 cm inlet is arbitrary. There is little variation of Hg(0) with height on the meteorological tower. Displaying all of the Hg(0) data (i.e., at the three inlets of the meteorological tower) makes this Figure difficult to read.

A Figure displaying the annual variation of Hg(0) concentrations in the snow interstitial air collected at the various inlets of the two snow towers has been added in the revised manuscript:

[Figure]

"Figure 10: Annual variation of hourly-averaged Hg(0) concentrations (in ng/m³) in the snow interstitial air collected at the various inlets of the two snow towers: a) snow tower #1, b) snow tower #2. Note that we regularly experienced technical problems on snow tower #2 leading to missing values."

**- Fig. 4: I only see error bars on a few points. Are these the only ones with replicates? How many replicates were done in those cases? A line or two in the caption to explain this would be helpful.**

All samples were analyzed in replicates of three. Standard error is most of the time smaller than the width of the dot explaining why you only see error bars on a few points. This has been clarified in the caption of the revised manuscript:

"Figure 5: Total mercury concentration (ng/L), along with standard errors, in surface snow samples collected weekly at Concordia Station from February 2013 to January 2014. Dark period (winter) highlighted in grey, sunlit period highlighted in yellow. Total mercury concentrations were elevated (up to 74 ng/L) in November-December 2013 (summer). All samples were analyzed in replicates of three. Standard errors are frequently smaller than the width of the dots."

**- Section 3.2: You refer to "continuous" oxidation in the summer. What do you mean by that? It is clear there is net oxidation, but I am not sure you have shown it is continuous.**

Yes, indeed. The term "continuous oxidation" has been removed throughout the revised manuscript.

**- Fig. 7: This figure is a nice attempt to visualize the data, but it is rather confusing. Does the shading represent some sort of unspecified interpolation? Do the top boxes represent the met tower and the bottom the snow tower? In that case, why do the top boxes extend down below zero and the bottom ones not go up to 50 cm? If not, why don't the top and bottom agree within the overlap region?**

We agree that this figure, as it is, might be confusing. It has been modified in the revised version of the manuscript:

[Figure]

"Figure 12: Hourly (local time) mean atmospheric and interstitial air Hg(0) concentrations in a) summer, and b) spring/fall. The vertical axis is the height of measurement relative to the snow surface (in cm). Color contours show Hg(0) concentrations (in ng/m$^3$). Concentrations at 25, 210, and 1070 cm above the snow surface were acquired on the meteorological tower while concentrations at 50, 10, -10, -30, -50 cm, and -70 cm were collected on snow tower #1. Data were cubic spline interpolated using software R."

**- Fig. 8: I think this figure is not crucial to the paper, since it is only being used to support a suggested mechanism for a single extreme value. I think you can make that suggestion without an additional figure, though it is up to your discretion.**

We agree. This Figure has been removed in the revised manuscript.

**- Section 3.3.2: You mention a shift from oxidation to reduction at the beginning of winter, but it would be very helpful to see the time series of Hg(0) at depth (as mentioned above) – is it a sudden drop to a stable "winter" value, or is there a longer trend over the winter to accompany the atmospheric decline?**

As mentioned above, a Figure displaying the annual variation of Hg(0) concentrations in the snow interstitial air collected at the various inlets of the two snow towers has been added in the revised manuscript. It is a sudden drop and not a longer trend over the winter to accompany the atmospheric decline.

**- Section 3.4.1: Your summary is a bit confusing (ll. 339-343). I think you are saying by "continuous" that (i) has a week diurnal cycle and by "important" that (iii) has a strong diurnal cycle, resulting in the observed concentration pattern. Can you say this more clearly? I'm not sure what "important" means in (ii). Clearly it is important to the surface snow THg, but I'm not sure how this is related to the Hg(0) diurnal cycle.**

The summary has been modified in the revised manuscript:

"In summary, the observed summertime Hg(0) diurnal cycle in ambient air might be due to a combination of factors: i)  the intense oxidation of Hg(0) in ambient air due to the high oxidative capacity on the plateau – as evidenced by low mean Hg(0) concentrations (see section 3.2.1.a), ii)  subsequent Hg(II) deposition onto snowpack – as evidenced by elevated total mercury levels in surface snow samples (see section 3.2.1.a), and iii)  emission of Hg(0) from the snowpack during convective hours. Fig. 8 summarizes the processes that govern mercury exchange at the air/snow interface. Redox processes occurring within the snowpack are discussed in details in section 3.2."

**- Section 3.4.2: Where is this Hg(0) building up from? Presumably the snow, but it's not mentioned. Why is the fall concentration higher than spring? The reservoir of Hg in the summer snow?**

Yes indeed, we believe that Hg(0) is building up due to emissions from the snowpack. This has been clarified in the revised manuscript: "We believe that the shallow boundary layer could cause Hg(0) concentrations in ambient air to build up to where they exceeded levels recorded at lower latitudes in the Southern Hemisphere because Hg(0) – emitted from the snowpack – was dispersed into a reduced volume of air, limiting the dilution".

The fall concentration is higher than spring indeed likely due to the reservoir of mercury in the summer snow. It should be noted that in both 2012 and 2013 mercury depletion events occurred at the end of summer likely leading to Hg(II) deposition on the snowpack.

**- Fig. 9g: What inlet is the Hg(0) cycle from?**

This has been added in the caption of the revised manuscript:

"Figure 7: Hourly (local time) mean variation, along with the 95% confidence interval for the mean, of: **a)** Hg(0) concentration (in ng/m$^3$) at 25 cm above the snow surface, **b)** downwelling shortwave (SW) radiation (in W/m$^2$) according to the MAR model simulations, **c)** temperature (in °C) at 3 m above the snow surface, **d)** wind speed at 3 m above the snow surface (in m/s), **e)** planetary boundary layer (PBL) height (in m) according to the MAR model simulations, **f)** friction velocity ($u_*$, in m/s), and **g)** Eddy diffusivity ($K$, in m$^2$/s) in summer (red), fall (green), winter (blue), and spring (purple). Note that the hourly mean variation of Hg(0) concentration in summer is similar at the three inlets of the meteorological tower".

**- Section 3.5: It's a bit odd to refer to Fig. 10e first. I suggest you rearrange the figure to make this 10a.**

We agree. The Figure has been rearranged in the revised manuscript.

**- Section 3.6: (a) Given you have a single winter of data, and the decline is not seen at the other stations, can you eliminate instrument drift (e.g. trap poisoning)? Were the external calibrations before and after the winter consistent?**

The instrument failed in 2014 but the decreasing trend has also been observed at Concordia Station in 2015. The data are presented in a paper that will soon be submitted in *Atmospheric Chemistry and Physics* (Angot et al., in preparation). Additionally, the decreasing trend has also been observed at Dumont d'Urville and data are presented in a companion paper (Angot et al., 2016). These results give us confidence that the decline is not due to an instrument drift.

**(b) Why do you not include dry deposition of Hg(0) as a possible mechanism for this decrease? Given the low BL, what deposition flux would be needed to remove the observed amount of Hg(0)? How does this compare to other observations/calculations (Cobbett et al 2007, Zhang et al 2009)? I think it's quite similar. It would also account for the gradient in the decrease (3.6.2).**

You are absolutely right. Dry deposition of Hg(0) is a possible mechanism for this decrease. This has been added in the revised manuscript:

"The observed declining trend could also be attributed to the dry deposition of Hg(0) onto the snowpack. The dry deposition velocity is defined as follows (Joffre, 1988), as the ratio between the deposition flux $F$ (ng/m$^2$/s) and the concentration $C$ (ng/m$^3$):

$$v_d = \frac{F}{C} \qquad (2)$$

Denoting the height of the boundary layer h and the Hg(0) concentration at the beginning of winter $C_0$, the evolution of the concentration versus time is thus given by the following ordinary differential equation:

$$C = C_0\, e^{-(v_d/h)t} \qquad (3)$$

During winter (t = 107 days), the Hg(0) concentration gradually decreased from $C_0 \sim 1.03$ ng/m$^3$ to $C \sim 0.73$ ng/m$^3$ at 25 cm above the snowpack, in a mixing layer of 25 m high. According to Equation (3) the associated dry deposition velocity is 9.3 10$^{-5}$ cm/s. This result is in very good agreement with dry deposition velocities reported for Hg(0) over snow (Cobbett et al., 2007; Zhang et al., 2009)."

**(c) Speaking of which, you don't report in 3.1 if there are any seasonal differences in the three met tower inlets. Your discussion of the winter data suggests there would be.**

This has been added in the revised manuscript:

Lines 209-213: "No significant difference was observed between annual averages of Hg(0) concentrations measured at the three inlets of the meteorological tower in 2013 (*p* value = 3.1.10$^{-14}$, Mann-Whitney test). It should be noted that Hg(0) concentrations at the three inlets were significantly different in winter only (see section 3.1.4)."

Lines 372-376: "In 2013, the height of measurement had a significant influence on the decline over time of Hg(0) concentrations (ANCOVA test, *p* value < 0.05), with a steeper decrease at 25 cm than at 1070 cm. Additionally, wintertime Hg(0) concentrations were significantly lower at 25 cm than at 1070 cm (*p* value < 0.05, Mann-Whitney test)."

**(d) This section is poorly organized. I suggest removing the sub-sections since you basically discount the gas-phase reaction without doing so explicitly.**

Sub-sections have been removed in the revised version of the manuscript.

**(e) l. 402: Why don't you report your O3 data instead of (or as well as) referring to another paper? Also, are there O3 data at Troll or Neumayer that suggest that a winter reaction with O3 would not also happen there?**

The annual variation of O$_3$ measurements in 2012 and 2013 has been added in Figure 3 of the revised manuscript:

[Figure]

"Figure 3: Annual variation in 2012 and 2013 of a) hourly-averaged Hg(0) concentrations (in ng/m$^3$) at 500 cm and 25 cm above the snow surface in 2012 and 2013, respectively, b) downwelling shortwave (SW) radiation (in W/m$^2$), c) planetary boundary layer (PBL) height (in m), and d) ozone (O$_3$, daily mean in 2012 and hourly mean in 2013) mixing ratios (in ppbv). The vertical dashed lines represent seasonal boundaries."

The reason why the Hg(0) decline throughout winter does not occur at Troll or Neumayer is unclear. This could be due to meteorological conditions on the Antarctic plateau (e.g., temperature, relative humidity, boundary layer dynamics). Further research is clearly needed.

**- Section 3.7: A bit more detail about the observations that are attributed to transport from the plateau (ll. 457-462) would be helpful, as mentioned above.**

See response above.

**- Section 4: (a) I'd like to see a mention of the intriguing winter subsurface Hg(0) peak in here.**

This has been added in the conclusion of the revised manuscript:

"Additionally, Hg(0) concentrations increased with depth in the snow interstitial air in winter likely due to a dark reduction of Hg(II) species accumulated within the snowpack during the sunlit period."

**(b) Maybe change "heating" to "snowpack ventilation" or "ventilation and heating" in l. 479.**

This has been changed in the revised manuscript: "Summertime Hg(0) concentration in ambient air exhibited a pronounced diurnal cycle likely due to large emissions from the snowpack as a response to daytime snowpack ventilation."

**(c) In l. 481 I would change "likely" to "possibly"…. And do you really think gas-phase oxidation is even that likely? Your earlier discussion suggests not. You may need to add dry deposition of Hg(0) as well, depending what you find.**

Indeed, the decreasing trend observed in winter is most likely due to the dry deposition of Hg(0). This has been changed in the revised manuscript (abstract and conclusion).

**3. Specific comments**

**l. 32: change "never been observed" to "not been reported"**

Done.

**l. 46: change "contamination" to "contaminant"**

Done.

**l. 246: change "bound" to "bind"**

Done.

**l. 315: change "significant and daily" to "significant daily"**

Done.

**l. 316: change "all along" to "throughout"**

Done.

**l. 358: suggest changing "explosions" to "so-called 'bromine explosions'" to avoid leaving unfamiliar readers with the impression there are actual explosions.**

We agree. This has been corrected in the revised manuscript.

**l. 378: change "these depletions of Hg(0)" to "the depletions of Hg(0) reported here"**

Done.

induced reactivity of atmospheric mercury during the sunlit period. The mechanisms which
cause the seasonal variation of Hg(0) concentrations are discussed in the following sections.

**3.1.1  Spring**

[revised manuscript text omitted]
 oxidative capacity on the plateau – as evidenced by low mean Hg(0) concentrations (see section 3.2.1.a), ii) important subsequent Hg(II) deposition onto snowpack – as evidenced by elevated total mercury levels in surface snow samples (see section 3.2.1.a), and iii) important emission of Hg(0) from the snowpack during convective hours. Fig. 8 summarizes the processes that govern mercury exchange at the air/snow interface. Redox processes occurring within the snowpack are discussed in details in section 3.2.

**3.1.3 Fall**

In fall, Hg(0) concentrations in ambient air no longer peaked around midday (see Fig. 7a) and were in average 67% higher than during the summer, exceeding levels recorded at lower latitudes in the Southern Hemisphere (Slemr et al., 2015). At this period of the year, the boundary layer lowered to ~ 50 m in average and no longer exhibited a pronounced diurnal cycle (see Figs. 3c and 7e). We believe that the shallow boundary layer could cause Hg(0) concentrations in ambient air to build up to where they exceeded levels recorded at lower latitudes in the Southern Hemisphere because Hg(0) – emitted from the snowpack – was dispersed into a reduced volume of air, limiting the dilution. Similarly, NO$_x$ mixing ratios are enhanced when the boundary layer is shallow (Neff et al., 2008; Frey et al., 2013). Elevated Hg(0) concentrations were also likely favored by the fact that oxidation in ambient air was weaker under lower solar radiation.

**3.1.4 Winter**

While stable concentrations were expected in winter given the absence of photochemistry, our observations reveal a 20 to 30% decrease of atmospheric Hg(0) concentrations from May to mid-August (see Fig. 3a). Conversely, Hg(0) concentrations remained stable at Neumayer and Troll from late fall through winter (Ebinghaus et al., 2002; Pfaffhuber et al., 2012). This decreasing trend observed in winter might be due to several mechanisms, including gas-phase oxidation, and heterogeneous reactions, or dry deposition of Hg(0).

Several studies suggested the involvement of nitrate radicals in the night-time oxidation of Hg(0) (Mao and Talbot, 2012; Peleg et al., 2015). However, as previously mentioned, Dibble et al. (2012) indicated that $NO_3$ binds Hg(0) too weakly to initiate its oxidation in the gas phase. Another potential oxidant is $O_3$, with this reactant reaching a maximum in the winter (see Fig. 3d). However, according to some theoretical studies (e.g., Hynes et al., 2009), reaction (R1) is unlikely to proceed as a homogeneous reaction. Several experimental studies confirmed the major product of reaction (R1) to be solid mercuric oxide, HgO (s) and not HgO (g) (e.g., Pal and Ariya, 2004; Ariya et al., 2009), suggesting that pure gas phase oxidation of elemental mercury by $O_3$ may not occur in the atmosphere. However, Calvert and Lindberg (2005) proposed an alternative mechanism that would make this reaction potentially viable in the atmosphere (Subir et al., 2011). The reaction may start with the formation of a metastable $HgO_3$ (g) molecule which then decomposes to OHgOO (g) and thereafter transforms to HgO (s) and $O_2$ (g).

$$Hg(0)\ (g) + O_3\ (g) \rightarrow HgO\ (g) + O_2\ (g)\ (R1)$$

As suggested by Subir et al. (2011), the influence of heterogeneous surfaces of water droplets, snow, ice and aerosols should be taken into account when attempting to describe mercury chemistry in the atmosphere. O'Concubhair et al. (2012) showed that freezing an acidic solution containing nitrite or hydrogen peroxide can oxidize dissolved gaseous mercury in the dark. Nitrous acid and hydrogen peroxide are present on the Antarctic plateau (Huey et al., 2004; Hutterli et al., 2004). As suggested by Dommergue et al. (2012), similar processes could occur in the snow or on surface hoar at Concordia Station in winter. In 2013, the height of measurement had a significant influence on the decline over time of Hg(0) concentrations (ANCOVA test, $p$ value $< 0.05$), with a steeper decrease at 25 cm than at 1070 cm. Additionally, wintertime Hg(0) concentrations were significantly lower at 25 cm than at 1070 cm ($p$ value $< 0.05$, Mann-Whitney test). These results suggest that snowpack may act as a sink for mercury, enhancing the deposition rate due to heterogeneous reactions, through absorption of oxidation products, and/or physical sorption/condensation of Hg(0) on surface snow.

The observed declining trend could also be attributed to the dry deposition of Hg(0) onto the snowpack. The dry deposition velocity is defined as follows (Joffre, 1988), as the ratio between the deposition flux $F$ (ng/m$^2$/s) and the concentration $C$ (ng/m$^3$):

$$v_d = \frac{F}{C} \qquad (2)$$

Denoting the height of the boundary layer h and the Hg(0) concentration at the beginning of
winter $C_0$, the evolution of the concentration versus time is thus given by the following
ordinary differential equation:

$$C = C_0\, e^{-(v_d/h)t} \qquad (3)$$

During winter (t = 107 days), the Hg(0) concentration gradually decreased from $C_0$ ~ 1.03
ng/m$^3$ to $C$ ~ 0.73 ng/m$^3$ at 25 cm above the snowpack, in a mixing layer of 25 m high.
According to Equation (3) the associated dry deposition velocity is 9.3 10$^{-5}$ cm/s. This result
is in very good agreement with dry deposition velocities reported for Hg(0) over snow
(Cobbett et al., 2007;  Zhang et al., 2009).

[revised manuscript text omitted]
. Additionally, Hg(0) concentrations increased with depth in the snow interstitial air in winter likely due to a dark reduction of Hg(II) species accumulated within the snowpack during the sunlit period. Finally, we reveal the occurrence of multi-day to weeklong depletion events of Hg(0) in ambient air in summer, that are not associated with depletion of $O_3$, and likely result from a stagnation of air masses on the plateau triggering an accumulation of oxidants in the shallow boundary layer. This behaviour is radically different from what is usually observed in the Arctic where only mercury depletion events that were associated with $O_3$ depletion (and with a Hg(0)/$O_3$ correlation) have been highlighted so far. According to observations at coastal Antarctic stations (see section 4), the reactivity observed at Concordia Station can be transported at a continental scale by strong katabatic winds. Our understanding of the atmospheric mercury chemistry on the Antarctic plateau is currently limited by the lack of continuous halogens measurements. Our findings point out new directions for future kinetic, observational, and modeling studies.

**Acknowledgements**

Mercury data reported in this paper are available upon request at http://sdi.iia.cnr.it/geoint/publicpage/GMOS/gmos_historical.zul. We thank A. Barbero and the rest of the overwintering crew: S. Aubin, C. Lenormant, and R. Jacob. We also gratefully acknowledge M. Barret for the development of a QA/QC software program, L. Bonato for the analysis of total mercury in surface snow samples, D. Liptzin for the calculation of the Obukhov length and friction velocity, M. Legrand for the 2012 ozone data, C. Genthon for the meteorological data, E. Vignon for helpful discussion, and B. Jourdain and X. Faïn for their help in the field. This work contributed to the EU-FP7 project Global Mercury Observation System (GMOS – www.gmos.eu) and has been supported by a grant from Labex OSUG@2020 (Investissements d'avenir – ANR10 LABX56), and the Institut Universitaire de France. BQ and PR acknowledge ANR CLIMSLIP and HAMSTRAD Program 910, respectively. Logistical and financial support was provided by the French Polar Institute IPEV (Program 1028, GMOstral and Program 1011, SUNITEDC), and a grant from the U.S. National Science Foundation (NSF, PLR#1142145). Computing resources for FLEXPART simulations were provided by the IPSL CICLAD/CLIMSERV mesocenter. Meteorological data were obtained thanks to LEFE/IMAGO programs CLAPA and GABLS4, IPEV program CALVA/1013, and Observatoire des Sciences de l'Univers de Grenoble (GLACIOCLIM observatory).

[Figure]

**Figure 1:** Map of Antarctica showing surface elevation (meters above sea level, m a.s.l) and the position of stations where atmospheric mercury measurements have been performed with modern on-line instruments. The black line shows the periphery of the high altitude plateau (> 2500 m a.s.l).

**a)**

[Figure]

**b)**

[Figure]

**Figure 2:** Photographs showing **a)** the meteorological tower with the three gas inlets (red arrows) at 1070 cm, 210 cm and 25 cm above the snow surface (photo credit: B. Jourdain), and **b)** one of the snow towers with the two sampling inlets above the snowpack at 50 and 10 cm (photo credit: D. Helmig).

[Figure]

**Figure 3:** Annual variation in 2012 and 2013 of **a)** hourly-averaged Hg(0) concentrations (in ng/m$^3$) at 500 cm and 25 cm above the snow surface in 2012 and 2013, respectively, **b)** downwelling shortwave (SW) radiation (in W/m$^2$), **c)** planetary boundary layer (PBL) height (in m), and **d)** ozone (O$_3$, daily mean in 2012 and hourly mean in 2013) mixing ratios (in ppbv). The vertical dashed lines represent seasonal boundaries.

[Figure]

**Figure 4:** Back trajectories for the 3 layers of altitude colored according to the potential emission sensitivity (PES, in seconds) **a)** on 11 September 2013, **b)** from 19 January to 8 February 2012, **c)** from 5 to 20 February 2013, **d)** on 10 February 2012, and **e)** on 22 February 2013. Note that PES in a particular grid cell is proportional to the particle residence time in that cell.

[Figure]

**Figure 5:** Total mercury concentration (ng/L), along with standard errors, in surface snow samples collected weekly at Concordia Station from February 2013 to January 2014. Dark period (winter) highlighted in grey, sunlit period highlighted in yellow. Total mercury concentrations were elevated (up to 74 ng/L) in November-December 2013 (summer). All samples were analyzed in replicates of three. Standard errors are frequently smaller than the width of the dots.

[Figure]

**Figure 6:** Top: January and February 2012 cycle of: **a)** hourly-averaged Hg(0) concentrations (in ng/m$^3$) at 500 cm above the snow surface, **b)** Integrated Water Vapor (IWV, kg/m$^2$), **c)** Temperature (in °C) at 10 m above ground level, and **d)** ozone (O$_3$, daily mean) mixing ratios (ppbv). Hg(0) was low from 19 January to 8 February (period highlighted in red) while O$_3$ showed no abnormal variability. Bottom: January and February 2013 cycle of: **e)** hourly-averaged Hg(0) concentrations (in ng/m$^3$) at 210 cm above the snow surface, **f)** Integrated Water Vapor (IWV, kg/m$^2$), **g)** Temperature (in °C) at 10 m above ground level, and **h)** ozone (O$_3$, hourly mean) mixing ratio (ppbv). Hg(0), IWV, and temperature were low from 5 to 20 February (period highlighted in red) while O$_3$ showed no abnormal variability. Note that Hg(0) concentrations exhibited the same pattern at the three inlets of the meteorological tower from 5 to 20 February 2013.

[Figure]

**Figure 7:** Hourly (local time) mean variation, along with the 95% confidence interval for the mean, of: **a)** Hg(0) concentration (in ng/m³) at 25 cm above the snow surface, **b)** downwelling shortwave (SW) radiation (in W/m²) according to the MAR model simulations, **c)** temperature (in °C) at 3 m above the snow surface, **d)** wind speed at 3 m above the snow surface (in m/s), **e)** planetary boundary layer (PBL) height (in m) according to the MAR model simulations, **f)** friction velocity ($u_*$, in m/s), and **g)** Eddy diffusivity ($K$, in m²/s) in summer (red), fall (green), winter (blue), and spring (purple). Note that the hourly mean variation of Hg(0) concentration in summer is similar at the three inlets of the meteorological tower.

[Figure]

**Figure 8:** Schematic diagram illustrating the processes that govern the Hg(0) budget at Concordia Station **a)** in summer under stable Planetary Boundary Layer (PBL) conditions, **b)** in summer under convective PBL conditions, **c)** in spring/fall, and **d)** in winter. In summer, Hg(0) is  intensely oxidized due to the high oxidative capacity of the boundary layer and a large amount of divalent mercury species deposit onto the snowpack. A fraction of deposited mercury can be reduced (the reducible pool, $Hg_r(II)$) in the upper layers of the snowpack and subsequently reemitted to the atmosphere as Hg(0). Hg(0) emission from the snowpack maximizes near midday likely due to increased ventilation as a response to daytime heating. Oxidation of Hg(0) dominates in the deeper layers of the snowpack likely leading to the formation of a Hg(II) reservoir. In spring/fall, the balance of reduction and oxidation processes within the upper layers of the snowpack differs from summertime: oxidation dominates during the day, reduction at night. In winter, Hg(0) is produced in the deeper layers of the snowpack likely as a result of the reduction of Hg(II) species accumulated during the sunlit period. Ambient Hg(0) concentrations exhibit a 20 to 30% decrease through winter possibly due to dry deposition of Hg(0).

[Figure]

**Figure 9:** Year 2012 wintertime record of: **a)** hourly-averaged Hg(0) concentrations (in ng/m$^3$) at 500 cm above the snow surface, **b)** Integrated Water Vapor (IWV, kg/m$^2$), and **c)** Temperature (T, °C) at 10 m above ground level. Hg(0), temperature, and IWV increased from June 12 to 15 (in red) suggesting transport of moist and warm air masses originating from lower latitudes.

[Figure]

**Figure 10:** Annual variation of hourly-averaged Hg(0) concentrations (in ng/m$^3$) in the snow interstitial air collected at the various inlets of the two snow towers: **a)** snow tower #1, **b)** snow tower #2. Note that we regularly experienced technical problems on snow tower #2 leading to missing values.

[Figure]

**Figure 11:** Mean Hg(0) concentration (ng/m$^3$) measured at various heights above and below the snow surface (cm) at Concordia Station in summer (yellow), spring/fall (purple), and winter (dark blue) The horizontal light blue dashed line represents the snow surface.

[Figure]

**Figure 12:** Hourly (local time) mean atmospheric and interstitial air Hg(0) concentrations in **a)** summer, and **b)** spring/fall. The vertical axis is the height of measurement relative to the snow surface (in cm). Color contours show Hg(0) concentrations (in ng/m$^3$). Concentrations at 25, 210, and 1070 cm above the snow surface were acquired on the meteorological tower while concentrations at 50, 10, -10, -30, -50 cm, and -70 cm were collected on snow tower #1. Data were cubic spline interpolated using software R.